# Fast and Expressive Multi-Byte Prediction with Probabilistic Circuits

**Andreas Grivas** [1]   **Lorenzo Loconte** [1]   **Emile van Krieken** [2]   **Piotr Nawrot** [1]   **Yu Zhao** [1]   **Euan Wielewski** [3]
**Pasquale Minervini** [1 4]   **Edoardo Ponti** [1] [*]   **Antonio Vergari** [1] [*]

## Abstract

Multi-token prediction (MTP) is a prominent strategy to significantly speed up generation in large language models (LLMs), especially in byte-level LLMs, which are tokeniser-free but prohibitively slow. However, many existing MTP methods either assume independence between future tokens, sacrificing expressiveness, or generate tokens one at a time within the window, increasing latency. In this work, we investigate the trade-off between expressiveness and latency in MTP within the framework of probabilistic circuits (PCs). Our framework, MTPC, allows one to explore different ways to encode the *joint* distributions over future tokens by selecting circuit architectures, generalising classical models such as (hierarchical) mixture models, hidden Markov models, and tensor networks. We show the efficacy of MTPC by retrofitting existing byte-level LLMs, such as EvaByte, and byte-fied subword models, such as Llama3.2 3B. Our experiments show that, when combined with speculative decoding, MTPC substantially speeds up generation compared to MTP with independence assumptions, while guaranteeing to retain the performance of the original verifier LLM. We also rigorously study the optimal trade-off between expressiveness and latency when exploring the possible parameterisations of MTPC, such as PC architectures and partial layer sharing between the verifier and draft LLMs.[1]

## 1. Introduction

Autoregressive (AR) large language models (LLMs) can only perform single-token prediction (STP) as they generate one token at a time, incurring significantly high latency, en-

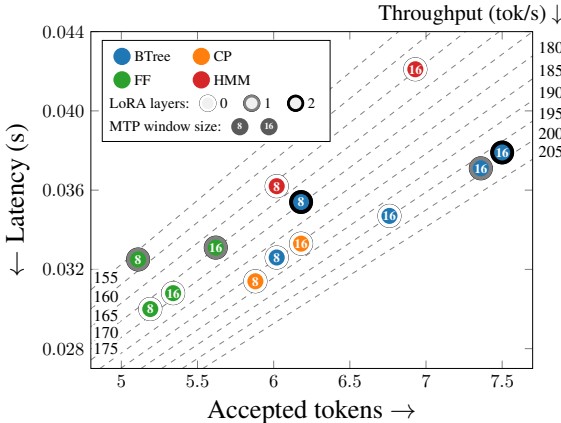

*Figure 1.* **MTPC allows us to trade-off efficiency (latency) and expressiveness (token acceptance) with different MTP designs** by choosing 1) the probabilistic circuit (PC) architecture (FF, CP, HMM, BTree); 2) the number of layers shared between draft and verifier models in self-speculative sampling. Dotted lines indicate iso-throughput (bytes/sec) regions, highlighting designs such as BTree for 16 bytes and 1 LoRA layer that achieve the best throughput for our retrofitted EvaByte-MTP model (Sec. 4).

ergy demand, and deployment costs. This problem affects subword models, but is dramatically exacerbated in byte-level ones (Minixhofer et al., 2025; Wang et al., 2024; Yu et al., 2023, *inter alia*) as many more decoding steps are required to generate a text of the same length. Among possible alternatives to speed up generation (Ankner et al., 2024; DeepSeek-AI et al., 2024; Nawrot et al., 2023; Pagnoni et al., 2025; Łańcucki et al., 2026), multi-token prediction (MTP) stands out as it promises to predict a window of multiple tokens *all at once*, may they be subwords (Gloeckle et al., 2024; Cai et al., 2024) or bytes (Gloeckle et al., 2024; Zheng et al., 2025). As such, MTP LLMs can achieve a significantly higher throughput than STP ones, as they decrease the number of passes through the LLM to generate the same text. While recent works on MTP focuses on subword models (Cai et al., 2024; Li et al., 2024), MTP can improve throughput the most in byte-level models (Pagnoni et al., 2025; Wang et al., 2024) due to their increased computational overhead. If one could model large sequences

---

[*]Joint last authors.   [1]School of Informatics, University of Edinburgh, UK [2]Vrije Universiteit Amsterdam, NL [3]NatWest Group [4]Miniml.AI. Correspondence to: Andreas Grivas <agrivas@ed.ac.uk>.

*Proceedings of the $43^{rd}$ International Conference on Machine Learning*, Seoul, South Korea. PMLR 306, 2026. Copyright 2026 by the author(s).

[1]Code and models available at github.com/april-tools/mtpc.

of future bytes, byte-level LLMs could become more mainstream as they already obviate many limitations of subword tokenizers, such as uneven efficiency (Ahia et al., 2023; Dagan et al., 2024), lack of interoperability (Minixhofer et al., 2026), and vulnerabilities (Rumbelow & Watkins, 2023; Land & Bartolo, 2024; Geiping et al., 2024).

However, modelling the joint distribution over many future bytes as tokens in a window is challenging, as it requires balancing *expressiveness*, i.e., representing all the dependencies between bytes, and *efficiency*, i.e., minimising latency. Many existing MTP approaches favour the latter by making an unrealistic assumption: they consider all future tokens to be independent (Zheng et al., 2025; Cai et al., 2024; Gloeckle et al., 2024). This clearly comes at the expense of expressiveness (Ankner et al., 2024; Wertheimer et al., 2024), as the choice of a byte for a position within the window cannot influence the probability of the others.

For example, consider the prompt: "*Name a capital of South Africa*", where *Cape Town* and *Pretoria* are equally likely completions. A byte-level MTP model with independence assumptions over an 8-token window could return *Cretoria* as an argmax, because replacing *P* with *C* cannot change the probability of other tokens. More concerningly, a number of "byte-salad" continuations, such as *Crptoria*, *Crpt ria* and *Crpt roa*, can also have high probability, despite having almost zero probability under the STP model. Most importantly, the number of these erroneous continuations grows exponentially with respect to the MTP window size, making the independence assumption problematic.

Perhaps the dominant paradigm for relaxing independence assumptions in MTP are shallow AR heads (Ankner et al., 2024; Li et al., 2024; DeepSeek-AI et al., 2024). However, these heads retain autoregressive generation within the prediction window, which we view as closer in spirit to accelerated AR decoding than to joint MTP. Importantly for our purposes, we seek methods that satisfy the following two *desiderata*: a) they yield a joint distribution over the token window that can be sampled from in parallel, and b) they expose the dependency structure of the joint as a modelling choice. Shallow AR heads satisfy neither. Moreover, while Hydra (Ankner et al., 2024) and EAGLE (Li et al., 2024) are state-of-the-art for subword-level MTP, their reported gains do not necessarily translate to byte-level models; a direct empirical comparison is orthogonal to our contribution and faces several technical obstacles, see Appendix D.

Satisfying our desiderata, Basharin et al. (2025) introduced dependencies into MTP using a mixture over the future token probabilities, but a single mixture can only add limited expressiveness. Crucially, understanding how to increase expressiveness while optimally trading off efficiency in a systematic way is still an open question. We address this gap by proposing an MTP framework based on probabilistic circuits (PCs; Choi et al., 2020; Vergari et al., 2021), which we name MTPC. MTPC uses PCs to parameterise the joint distribution over future tokens as tractable computational graphs that encode hierarchical mixture models. As such, MTPC offers a way to systematically navigate the spectrum of MTP architectural variants, encompassing fully factorised models (Zheng et al., 2025; Cai et al., 2024; Gloeckle et al., 2024), shallow mixtures (Basharin et al., 2025), but also more expressive parameterisations: hidden Markov models (HMMs) and binary tree factorisations (BTrees), which are novel for MTP.

Moreover, in contrast to previous work on MTP (Zheng et al., 2025; Cai et al., 2024), MTPC guarantees we match the quality of an AR LLM via speculative decoding (Leviathan et al., 2022; Chen et al., 2023; Stern et al., 2018; Xia et al., 2024). For greedy decoding, the output is guaranteed to be identical to the AR LLM; for sampling, it matches in expectation. This shows that the throughput cost of the guarantee is not as large as previously suggested. We achieve this by sharing the LLM backbone for the draft and verifier models for different numbers of layers, highlighting how this creates *a second dimension to trade-off expressiveness* (as hidden representations between draft and verifier are allowed to differ) *and latency* (as each non-shared layer requires separate forward passes). We illustrate the two trade-offs at the core of MTPC in Fig. 1.

In summary, we make the following contributions: **C1)** we introduce MTPC, a fast MTP framework based on PCs that relaxes the independence assumptions of previous work and generalises tensor decomposition methods (Basharin et al., 2025); **C2)** we rigorously identify trade-offs between the number of accepted tokens in speculative decoding and the latency of generation, based on different choices of PC architectures and partial layer sharing; **C3)** we empirically demonstrate the effectiveness of MTPC by repurposing two byte-level LLMs, EvaByte (Zheng et al., 2025) and Llama 3.2 3B Byte (Minixhofer et al., 2025), into our framework. We find that MTPC substantially increases the throughput of EvaByte by $5.15\times$ and that of Llama by $2.24\times$ with respect to AR generation and both by $1.17\times$ with respect to MTP with independence assumptions, by supporting MTP windows of up to 16 bytes.

## 2. Speeding up Generation with MTP and Speculative Decoding

We now introduce MTP and speculative decoding, the two main ingredients of MTPC.[2] The former accelerates generation from the target LLM; the latter guarantees that its generation quality is preserved.

---

[2]We adapt notation from the tensor and circuit literature (Loconte et al., 2025a); see also Appendix A.

**MTP.** A classical STP LLM encodes a distribution over sequences of tokens $\{\mathbf{x}_t\}$ defined over a vocabulary $\mathcal{V}$ as $\prod_t p(x_{t+1} \mid \mathbf{x}_{\leq t})$, where $\mathbf{x}_{\leq t}$ is the context, *i.e.* the observed tokens at timestep $t$. MTP (Gloeckle et al., 2024) aims to extend an STP LLM that predicts a single token at a time through $p(x_{t+1} \mid \mathbf{x}_{\leq t})$, to an MTP model, $q_{\boldsymbol{\theta}}$, that models the *joint* probability of a window of $n$ future tokens and generates them *simultaneously*, i.e. we model

$$q_{\boldsymbol{\theta}}(x_{t+1}, x_{t+2}, \ldots, x_{t+n} \mid \mathbf{x}_{\leq t}), \qquad (1)$$

where $\boldsymbol{\theta}$ denotes a given parameterisation for the joint.[3] The first dimension to trade-off expressiveness and efficiency in MTP pertains to compactly representing $q_{\boldsymbol{\theta}}$. Unlike for $p(x_{t+1} \mid \mathbf{x}_{\leq t})$, we would need to store more than a vector of logits $\mathbf{a} \in \bar{\mathbb{R}}^v$ of a single univariate categorical distribution for a vocabulary size $v = |\mathcal{V}|$ for every timestep $t$. The most expressive, but least efficient way to do so, would be to store an $n$-dimensional tensor $\boldsymbol{\mathcal{A}} \in \bar{\mathbb{R}}^{v^{(1)} \times \ldots \times v^{(n)}}$ of logits having $v^n$ entries, but this scales exponentially in $n$. Next, we review past attempts to avoid storing $\boldsymbol{\mathcal{A}}$ explicitly.

**Fully factorised.** A common way to boost efficiency is to assume all $n$ future tokens are independent (Zheng et al., 2025; Cai et al., 2024; Gloeckle et al., 2024) and factorise the distribution $q_{\boldsymbol{\theta}}$ in Eq. (1) as

$$\prod_{i=1}^{n} q_{\boldsymbol{\theta}}(x_{t+i} \mid \mathbf{x}_{\leq t}). \qquad \text{(FF)}$$

This comes with the benefit that one needs to store only $n$ probability vectors, each of dimension $v$, to represent the joint distribution in Eq. (1). However, as already discussed in the introduction, this severely limits the model's expressiveness (Ankner et al., 2024; Wertheimer et al., 2024).

**Canonical polyadic (CP) factorisation.** Dependencies between future tokens can be recovered by introducing explicit latent variables (Lee et al., 2018). To this end, Basharin et al. (2025) propose to factorise Eq. (1) via an $r$-rank CP decomposition. A CP decomposition introduces one discrete latent variable, $z$, that encodes a mixture of $r$ fully-factorised components, rewriting Eq. (1) as

$$\sum_{z=1}^{r} q(z \mid \mathbf{x}_{\leq t}) \prod_{i=1}^{n} q_{\boldsymbol{\theta}}(x_{t+i} \mid z, \mathbf{x}_{\leq t}), \qquad \text{(CP)}$$

where $q(z \mid \mathbf{x}_{\leq t})$ are the mixture coefficients.[4] Before showing how we can generalize both FF and CP MTP with probabilistic circuits, we review how to ensure MTP models match the quality of a given STP model.

---

[3]We drop the index $t$ from $\boldsymbol{\theta}_t$ for readability when not needed.

[4]Basharin et al. (2025) calls CP a mixture of experts (MoE), but we note this is incorrect as the weights $\omega_j$ do not depend on future tokens, but only on past ones. Moreover, while they argue that training CP is challenging and requires insights from the MoE literature, we can train them as well as deeper mixture variants easily without MoE-tailored losses (see Sec. 4).

**Speculative decoding** (Stern et al., 2018; Leviathan et al., 2022; Chen et al., 2023; Xia et al., 2024) can be combined with MTP to speed up generation while guaranteeing no loss in generation quality. Given a target STP LLM that we wish to accelerate, speculative decoding consists of a cycle of two steps: 1) *drafting*, where a cheaper MTP draft model generates $n$ future tokens, and 2) *verification*, where the target STP model accepts or rejects the generated tokens in parallel according to a pre-defined consistency criterion. The closer the distributions of the draft and verifier are, the more often 'speculated' tokens are accepted (Leviathan et al., 2022; Sun et al., 2023), speeding up generation. With speculative decoding, we quantify the trade-off between expressiveness and efficiency in MTP models as their *throughput*:

$$\text{throughput (tok/s)} = \frac{\text{number of generated tokens}}{\text{latency (s)}}, \quad (2)$$

where the numerator is approximately the number of accepted tokens and the denominator captures the wall-clock cost of generating the tokens (see Sec. 4.2 for details). While previous work such as Basharin et al. (2025) reports accepted token counts and latency separately, we highlight how both sides of the ratio in Eq. (2) are coupled, creating a spectrum. MTPC provides a systematic way to navigate this spectrum (see Fig. 1).

## 3. Probabilistic Circuits for MTP

The idea behind MTPCs is to further decompose the joint distribution in Eq. (1) into a deep computational graph encoding a hierarchical mixture model, called a *probabilistic circuit* (Secs. 3.1 and 3.2), and to parameterise it with LLM embeddings (Sec. 3.3).

### 3.1. Probabilistic Circuits

A ***circuit*** (Darwiche, 2003; Choi et al., 2020; Vergari et al., 2021), $c$, is a parameterised computational graph[5] over variables $\mathbf{X}$ encoding a function, $c(\mathbf{X})$, and comprises three kinds of computational units: *input*, *product*, and *sum* units. Each product or sum unit $n$ receives the outputs of other units as inputs, denoted with the set $\mathsf{in}(n)$. Each unit $n$ encodes a function, $c_n$, defined as: (i) $c_n(\mathsf{sc}(n); \phi)$ if $n$ is an input unit, where $c_n$ is a function parameterised by $\phi$ over variables $\mathsf{sc}(n) \subseteq \mathbf{X}$, called its *scope*; (ii) $\prod_{j \in \mathsf{in}(n)} c_j(\mathsf{sc}(j))$ if $n$ is a product unit; and (iii) $\sum_{j \in \mathsf{in}(n)} \omega_j c_j(\mathsf{sc}(j))$ if $n$ is a sum unit, with $\omega_j \in \mathbb{R}$ denoting the sum parameters. The scope of a product or sum unit $n$ is the union of the scopes of its inputs, i.e., $\mathsf{sc}(n) = \bigcup_{j \in \mathsf{in}(n)} \mathsf{sc}(j)$. Fig. 2 shows examples of circuits, where units of the same scope are grouped into (coloured) layers that can be easily parallelised on a GPU (Mari et al., 2023; Loconte et al., 2025a).

---

[5]In Fig. 2, the directionality of the edges is removed for readability, but it is assumed to be from inputs to outputs.

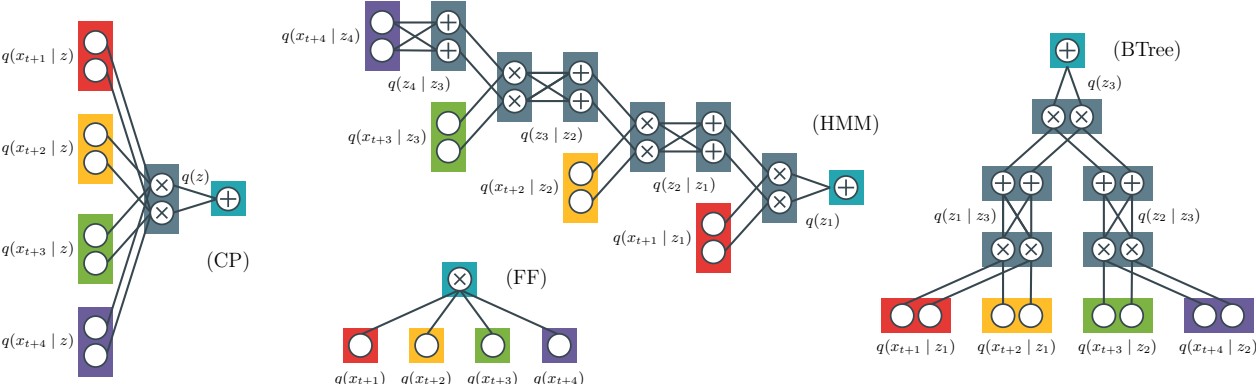

*Figure 2.* **PCs allow for modelling a spectrum of dependency structures** over sequences of tokens, as shown for the known FF and CP and the novel HMM and BTree MTP variants. Input units are grouped in coloured layers, one for each token, while sum and product layers encoding (hierarchies of) latent variable distributions are in grey. The output unit of each circuit (in blue) computes $q_{\theta}(x_{t+1}, \ldots, x_{t+n} \mid \mathbf{x}_{\leq t})$. In the figure we omit the dependency on the context $\mathbf{x}_{\leq t}$ for readability.

For MTPCs, we use ***probabilistic circuits*** (PCs), i.e., circuits modelling a joint distribution over random variables, in our case tokens. PCs encode Eq. (1) as

$$q_{\theta}(x_{t+1}, \ldots, x_{t+n} \mid \mathbf{x}_{\leq t}) = Z_{\theta_t}^{-1} \, c(x_{t+1}, \ldots, x_{t+n}; \theta_t), \quad (3)$$

where $\theta_t = \{\omega_t, \phi_t\}$ denote the set of circuit parameters, *i.e.*, all sum unit parameters $\omega_t$ and input unit parameterisations $\phi_t$ which depend on the context $\mathbf{x}_{\leq t}$; and $Z_{\theta_t}$ denotes the partition function of $c$, *i.e.*,

$$Z_{\theta_t} = \sum_{x_{t+1}, \ldots, x_{t+n} \in \mathcal{V}^n} c(x_{t+1}, \ldots, x_{t+n}; \theta_t). \quad (4)$$

Note that the PC architectures we are interested in are already normalised or always allow computing the partition function in a single feed-forward step (see Choi et al. (2020) and Appendix B.1). At the same time, we can easily sample from PCs in a single feedforward pass, as discussed in Appendix B.2. Crucially, within the framework of PCs, we can recover the FF and CP parameterisations for MTP and several other architectures that generalise tensor factorisations (Loconte et al., 2025a), each offering a different expressiveness-efficiency trade-off. We do so while abstracting away from each model's original formulation and obtain a unified way to parameterise MTP LLMs, as discussed next.

### 3.2. PC Architectures for MTP

**MTPC-FF.** Representing the commonly used FF MTP parameterisation as a PC is simple: we introduce $n$ input units, each parameterised by $\phi_i$, its corresponding token probabilities, and connect them all to a single product unit, as shown in Fig. 2 for a distribution over $n = 4$ tokens.

**MTPC-CP.** Similarly, we can easily encode a CP factorisation in a *shallow* PC by i) introducing $r$ input units for

each token (each parameterised by their own probabilities $\phi_{ij}$), then ii) multiplying them to retrieve the $r$ factorised mixture components, which we then iii) aggregate in a sum unit with weights $\omega_j = q(z_j \mid \mathbf{x}_{\leq t})$ (see also Proposition 1 in Loconte et al. (2025a)). Fig. 2 shows this construction for $n = 4$ and $r = 2$. This basic construction suggests that we can create deeper architectures by interleaving sum and product layers, while overparameterising each layer by increasing the number of units $r$ in it. Furthermore, implementing CP as a PC unlocks a faster sampling routine (Appendix B.2) than the one used in Basharin et al. (2025).

**MTPC-HMM.** As a further example of the expressiveness increase we get by generalising our approach to deeper PCs, we introduce a factorisation that realises a hidden Markov model (HMM), which better captures distant dependencies in the sequence by introducing *a sequence of latent variables*, in contrast to the single one in CP. More precisely, we define an HMM with $r$ hidden states and truncate its prediction window to $n$ steps into the future. We resort to an inhomogeneous HMM, *i.e.*, we use time-dependent transition matrices, as it is more expressive and worked better in our experiments (Appendix H). This simplifies Eq. (1) into:

$$\sum_{z_1=1}^{r} \cdots \sum_{z_n=1}^{r} q(z_1 \mid \mathbf{x}_{\leq t}) q_{\theta}(x_{t+1} \mid z_1, \mathbf{x}_{\leq t})$$
$$\times \prod_{i=2}^{n} q(z_i \mid z_{i-1}, \mathbf{x}_{\leq t}) q_{\theta}(x_{t+i} \mid z_i, \mathbf{x}_{\leq t}). \quad \text{(HMM)}$$

Fig. 2 illustrates the HMM parameterisation above represented as a circuit, comprising $n = 4$ stacked pairs of sum and product layers, where the parameters $\omega_i$ of the former are the transition probabilities $q(z_i \mid z_{i-1}, \mathbf{x}_{\leq t})$. Similarly to CP, we can increase $r$ to overparameterise the circuit with more input units per token and sum units overall, and hence increase expressiveness.

**MTPC-BTREE.** One drawback of the HMM parameterisation is the asymmetry of its computational graph, which

i) provides fewer latent variables for the early tokens, and ii) increases latency when predicting the last tokens due to its autoregressive token dependencies. To solve this, we build a PC whose structure resembles that of a binary tree (BTree), effectively encoding a *hierarchy of latent variables* or a tree tensor factorisation (Grasedyck, 2010; Cheng et al., 2019; Loconte et al., 2025a). This is done recursively: at each step $h$ of the hierarchy, given $n$ tokens, and a parent latent variable $Z_l$, we split the tokens into two sub-sequences $(x_{t+1}, \ldots, x_{t+\lfloor n/2 \rfloor -1})$ and $(x_{t+\lfloor n/2 \rfloor}, \ldots, x_{t+n})$, then factorise Eq. (1) as the mixture:

$$\sum_{z_h=1}^{r} q(z_h \mid z_l, \mathbf{x}_{\leq t})$$
$$\times q_{\boldsymbol{\theta}}(x_{t+1}, \ldots, x_{t+\lfloor n/2 \rfloor -1} \mid z_h, z_l, \mathbf{x}_{\leq t})$$
$$\times q_{\boldsymbol{\theta}}(x_{t+\lfloor n/2 \rfloor}, \ldots, x_{t+n} \mid z_h, z_l, \mathbf{x}_{\leq t}) \quad \text{(BTree)}$$

which corresponds to creating a sum unit whose weights are $q(z_h \mid z_l, \mathbf{x}_{\leq t})$ followed by products. We repeat the process while caching intermediate units until we reach the base case for $n = 1$, for which we create a layer of input units for the corresponding token. Fig. 2 illustrates the BTree circuit built in this way. Our experiments (Sec. 4) show that the BTree parameterisation obtains the optimal throughput by lowering the latency of the HMM, as it samples more latent variables and tokens in parallel, while achieving similar acceptance rates. In addition, our experiments show that the BTree parameterisation improves throughput on larger MTP window sizes as the balanced tree structure enables parallel sampling of tokens and latent variables.

### 3.3. Parameterising PCs with LLMs

Parameterising MTPCs requires two functions: an LLM that maps the context $\mathbf{x}_{\leq t} \in \mathcal{V}^t$ into contextual features, and a neural network head that maps the contextual features to the parameters of the circuit $\boldsymbol{\theta}_t$, realising a *neural conditional circuit* (Shao et al., 2020; 2022; Ahmed et al., 2022). To extract the contextual features $\mathbf{e}_t \in \mathbb{R}^d$, we use $\mathbf{e}_t = \text{LLM}_{\text{LoRA}(k)}(\mathbf{x}_{\leq t})$ where $\text{LLM}_{\text{LoRA}(k)} : \mathcal{V}^t \to \mathbb{R}^d$ is the STP backbone with LoRA (Hu et al., 2022) applied to the last $k \geq 0$ layers. As we will discuss in Sec. 4.4, the number of LoRA layers can impact throughput significantly. Given $\mathbf{e}_t$, we realise Eq. (3) by computing $\boldsymbol{\theta}_t = g_c(\mathbf{e}_t)$, where $g_c$ is a neural network head that outputs both the input unit parameters, $\boldsymbol{\phi}_t$, and the sum unit parameters, $\boldsymbol{\omega}_t$ (Sec. 3.1). Note that our parameterisation in MTPCs allows us to abstract from the actual structure of the circuit (i.e., FF, CP, HMM or BTree) and just focus on these two sets of tensorised parameters, as we discuss next.

**Input unit distributions.** All MTPCs produce joint distributions over token windows by combining categorical distributions over individual tokens (Fig. 2). We follow EvaByte (Zheng et al., 2025) and learn $n$ separate unembedding layers, one per window position. For models with mix-ture coefficients, we also learn one unembedding layer per mixture coefficient.[6] As such, instead of a single unembedding matrix mapping $\mathbb{R}^d \to \mathbb{R}^v$, we have an unembedding tensor $\boldsymbol{\mathcal{W}} \in \mathbb{R}^{n \times r \times v \times d}$, and compute the input distributions with the usual unembedding operation followed by softmax, i.e., $\boldsymbol{\phi}_{tij} = \text{softmax}(\boldsymbol{\mathcal{W}}_{ij}\mathbf{e}_t)$, where $i$ and $j$ index the position in the MTP window and the rank $r$.

**Sum unit parameters.** Instead of mapping embeddings to the vocabulary via $\boldsymbol{\mathcal{W}}$, we map to the rank of the sum unit via $\boldsymbol{\mathcal{R}} \in \mathbb{R}^{z \times r \times d}$, where $z$ is the number of sum units, $r$ is its rank, and $d$ the dimensionality of $\mathbf{e}_t$. We compute $\boldsymbol{\omega}_{ti} = \text{softmax}(\boldsymbol{\mathcal{R}}_i \mathbf{e}_t)$, where $i$ indexes the sum unit.

### 3.4. Speculative Decoding with MTPC

For MTPCs, we design an architecture that is *self-drafting* (Zhang et al., 2024b; Cai et al., 2024), i.e. where the draft and verifier models share the same LLM backbone. We use an MTP head (Cai et al., 2024; Ankner et al., 2024) augmented with our circuits to efficiently sample a draft, and an autoregressive STP head as the verifier. Optionally, we also keep a few final transformer layers separate in the two models by fine-tuning LoRA adapters for the draft model.

Unlike previous self-drafting MTP works (Cai et al., 2024; Ankner et al., 2024), we guarantee that the generated tokens are exactly the same (*greedy speculative decoding* (Stern et al., 2018)) or the same in expectation (*speculative sampling*) as those the autoregressive LLM would generate using *speculative decoding* (Leviathan et al., 2022; Chen et al., 2023), *i.e.*, we only generate the subset of drafted tokens accepted by our verifier. To keep latency low, we make only a single LLM call per speculative decoding cycle by re-using, where possible, the LLM backbone state computed by the verifier for the draft model. Sharing the state forfeits the bonus token of standard speculative decoding, except in the rare case where no tokens are accepted; see Alg. 2. The GPU memory overhead of speculative decoding with our most expressive PCs is only 10-15% more than MTPC-FF, see Fig. 5. Next, we report results for MTPC both for speculative sampling and greedy speculative decoding.

## 4. Retrofitting Byte-Level LLMs with MTPCs

We evaluate MTPC on the challenging tasks of speeding up byte-level LLMs. We speed-up two models, EvaByte 6.5B, which already has MTP capabilities, and Llama 3.2 3B (Byte), which does not and is a byte-fied version of the popular subword-level LLM. Detailed parameters of both models can be found in Appendix E. We implement our MTPCs variants in the `cirkit` library (The april Lab, 2024) and provide it in our supplementary materials.

---

[6]This is efficient even for PCs with high rank due to the small vocabulary size of byte-level LLMs.

**EvaByte 6.5B.** EvaByte (Zheng et al., 2025) is an open source, publicly available byte-level model that obtains results that are competitive to subword-level LLMs on benchmarks (Zheng et al., 2025), see Appendix E.1. Importantly, EvaByte has been pretrained as an MTP model with a prediction window of $n = 8$ bytes. As a result, it can already speed-up generation very effectively with greedy speculative decoding. However, its acceptance rate with speculative sampling is hampered by the unrealistic independence assumptions which we relax with MTPC. Moreover, we will see that EvaByte's pretraining enables us to successfully extend prediction to $n = 16$ tokens. Thus, we retrofit EvaByte-SFT (Zheng et al., 2025), which has been instruction fine-tuned on a data mix of Tülu 3 (Lambert et al., 2025), OpenCoder (Huang et al., 2025) stages one and two, and OpenHermes 2.5. We note that EvaByte's solid performance on benchmarks is obtained via Medusa-style lossy speculative decoding with the MTP head, which in the case of sampling comes with a loss in quality compared to EvaByte-STP ($n = 1$). We therefore set EvaByte-STP as the target model for speculative decoding to *accelerate generation without sacrificing generation quality*.

**Llama 3.2 3B (Byte).** We also retrofit a Llama3.2 3B model (Grattafiori et al., 2024) to show that our findings generalise. Since we focus on byte-level LLMs, we retrofit the byte-level version that has been distilled from the subword-level model in Minixhofer et al. (2026) while retaining most of Llama's downstream performance.[7] The byte-fied Llama3.2 3B model was fine-tuned on Tülu 3 with a context length of 2048. As opposed to EvaByte, the byte-fied version of Llama was not pretrained for multi-token prediction. As such, the boost we can get from MTPC is smaller than that for EvaByte, as it is more challenging to adapt it to long windows with LoRA adapters only on the last few layers.

**Draft models.** Each draft model comprises a backbone and an MTP head. We introduce two axes to control the draft model's expressiveness: i) more expressive circuit heads and ii) more LoRA layers in the draft backbone. For i) we combine the target's model backbone with one of our MTPCs heads, including our CP implementation and novel HMM and BTree heads to relax the independence assumptions of the FF model and increase expressiveness. We note that MTPC-CP with $r = 1$ is equivalent to MTPC-FF, as can be seen from Eq. (CP). For ii) we decouple the draft and target backbone by adding LoRA adapters to the last 1, 2 or 4 layers of the draft's backbone.

### 4.1. Training

We train 55 draft models in total. We improve throughput by making our MTP model's distribution as similar as possible to the target's in the simplest way: we instruction fine-tune

our MTP models on a similar data mix to that used for instruction fine-tuning the target.

**Training data.** We fine-tune on the Tülu 3 SFT mix dataset (Lambert et al., 2025) which contains 939,344 examples of user/assistant interactions on 18 tasks. We split the Tülu 3 dataset into training and validation and evaluate throughput on the unseen validation examples. We make sure all tasks are sampled by shuffling the training data before splitting. To make training possible on $2 \times 80$ GB GPUs, we limit the context length to 8192 bytes and filter out 34,067 examples which are longer. We split the remaining 905,277 examples into 99% train and 1% validation.

**Initialisation.** For EvaByte, we initialise our MTP heads from EvaByte-SFT in a way that guarantees that our EvaByte-MTP-CP is equivalent to EvaByte-MTP. This guarantees that we leverage previous training: all models start from the same loss and we smoothly move in parameter space from EvaByte-MTP to our more expressive EvaByte-MTP-CP, EvaByte-MTP-HMM and EvaByte-MTP-BTree.

**Loss.** We train our MTP models on the packed train split of Tülu 3 with a batch size of 256 sequences, or $\approx 2m$ tokens, which is what EvaByte used. We first train our MTP heads for 1 epoch (Sec. 4.3). Then we load the models and continue training for an additional epoch with LoRA (Sec. 4.4). We apply the target model's chat template and only train on the assistant's answers. We use overlapping prediction windows, as we need to be able to begin speculative decoding from any position during generation. We minimise the negative log-likelihood of the observed assistant outputs, see Eq. (5), where $N$ is the number of training sequences and $L$ is the sequence length for each token in the window.[8]

$$\mathcal{L}_j = -\frac{1}{N} \sum_{i=1}^{N} \frac{1}{V_{i,j}} \sum_{t=1}^{L} \log p_\theta(x_{t+j}^{(i)} \mid \mathbf{x}_{<t+j}^{(i)}) \quad (5)$$

This involves locally normalising the loss by the number of valid tokens for example $i$ and output $j$ in the MTP window, $V_{i,j}$. As in Cai et al. (2024), we apply exponential discounting to the loss $\mathcal{L}_j$ of token $j$ in the MTP window, *i.e.*, we minimise $\mathcal{L} = \sum_{j=1}^{n} \gamma^{j-1} \mathcal{L}_j$.[9] We use Adam (Kingma & Ba, 2015) with a fixed learning rate of $3 \times 10^{-4}$.

### 4.2. Metrics

To speed up LLM generation with speculative decoding, we need to balance **speed** and **expressiveness**. We measure speed using **mean latency per cycle** (⏱$\mu_{\text{lat}}$) and expressiveness via the **mean accepted tokens per cycle** (☑$\mu_{\text{acc}}$; Li et al., 2024), as defined below. Our goal is to increase **throughput**. We obtain a relative throughput speed-up

---

[8]Our loss over overlapping windows is a composite log-likelihood (Varin et al., 2011).

[9]We set $\gamma = 0.9$ instead of $\gamma = 0.8$ in the case of $n > 8$.

of one method over another by measuring their **wall-time speedup ratio** (Li et al., 2024; Cai et al., 2024). We use a batch size of 1 for all evaluations. Metrics for our main experiments are computed on the server-grade NVIDIA L40S GPU with complete results in Appendix K. We also run experiments on the desktop-grade NVIDIA RTX 3090 in Appendix L. We detail the metrics below.

**Mean latency** $\circlearrowleft\mu_{\text{lat}}$ is the average time of each speculative decoding cycle, *i.e.*, the time needed for the draft model to generate a candidate sequence and the verifier to choose which tokens to accept. $\circlearrowleft\mu_{\text{lat}}$ is higher for less efficient LLMs and MTP heads, and lower for more powerful GPUs.

**Mean accepted tokens** $\checkmark\mu_{\text{acc}}$ is the mean number of drafted tokens that are accepted by the target model per cycle. More expressive draft models will have higher acceptance as they will better approximate the target distribution. $\checkmark\mu_{\text{acc}}$ depends on the size of the MTP window, $n$, as we have $\checkmark\mu_{\text{acc}} \in [0, n]$.

**Mean throughput** $\mu_{\text{tok/s}}$ is the total number of generated tokens across all prompts divided by the total generation latency, excluding prefill (Eq. (2)). In the trade-off plots, we cannot directly visualise $\mu_{\text{tok/s}}$. Instead, we decompose throughput into the ratio $\checkmark\mu_{\text{acc}}$ / $\circlearrowleft\mu_{\text{lat}}$, which approximates Eq. (2) closely but not exactly: it is a ratio of per-cycle means rather than of totals, and accepted tokens slightly undercount generated tokens, as one token is still generated in the rare case where none are accepted (see Alg. 2).

**Wall-time speed-up ratio** is the relative speed-up of a proposed model compared to a baseline model, measured as the ratio of their $\mu_{\text{tok/s}}$. As baselines, we use autoregressive generation from the STP target model and MTP with independence assumptions.

### 4.3. MTPCs without Adapters

**RQ1**: Can we increase throughput by increasing the number of mixture components?

We begin with the simplest PC from our framework, MTPC-CP, which relaxes the independence assumption of the widely used MTPC-FF ($r = 1$) by increasing the number of mixture coefficients, $r$. MTPC-CP *increases throughput as it is more expressive than* MTPC-FF*, yet still very efficient.*

Table 1 highlights MTPC-CP's increase in expressivity as evidenced by the increase in $\checkmark\mu_{\text{acc}}$ as we increase $r$: MTPC-CP reaches $\checkmark\mu_{\text{acc}} = 6.02$ for $r = 128$, surpassing the MTPC-FF baseline by .83 for EvaByte with speculative sampling. At the same time, the $\circlearrowleft\mu_{\text{lat}}$ introduced by MTPC-CP increases only slightly as we increase $r$, because the forward pass cost of the MTPC-CP head is dominated by the expensive LLM calls. Nevertheless, as the increase in $\checkmark\mu_{\text{acc}}$ tails off for larger $r$, even the small $\circlearrowleft\mu_{\text{lat}}$ increase matters and we see the first of three cases in our experiments where the

*Table 1.* **Increasing the mixture components ($r$) increases the throughput** ($\mu_{\text{tok/s}}$) as seen for MTPC-CP ($n = 8$) over our baseline, EvaByte-MTP (FF) (in gray) where we report the mean $\pm$ std over three sets of 250 prompts of Tülu3 on an L40S GPU. MTPC-CP increases throughput: it has a larger acceptance ($\checkmark\mu_{\text{acc}}$) while latency ($\circlearrowleft\mu_{\text{lat}}$) increases slowly in $r$.

| | | $\circlearrowleft\mu_{\text{lat}} \downarrow$ | $\checkmark\mu_{\text{acc}} \uparrow$ | $\mu_{\text{tok/s}} \uparrow$ | $\max_{\text{tok/s}}$ |
|---|---|---|---|---|---|
| FF | 1 | **0.0299**$_{\pm 0.0003}$ | 5.19$_{\pm 0.05}$ | 176.3$_{\pm 0.9}$ | 297.55 |
| | 8 | 0.0314$_{\pm 0.0005}$ | 5.67$_{\pm 0.05}$ | 184.2$_{\pm 3.9}$ | 297.92 |
| | 16 | 0.0323$_{\pm 0.0020}$ | 5.79$_{\pm 0.08}$ | 183.4$_{\pm 13.3}$ | 290.71 |
| CP | 32 | 0.0314$_{\pm 0.0003}$ | 5.88$_{\pm 0.02}$ | **191.0**$_{\pm 1.6}$ | 287.88 |
| | 64 | 0.0327$_{\pm 0.0015}$ | 5.96$_{\pm 0.04}$ | 185.9$_{\pm 7.7}$ | 281.10 |
| | 128 | 0.0337$_{\pm 0.0001}$ | **6.02**$_{\pm 0.03}$ | 182.3$_{\pm 1.1}$ | 264.80 |

expressiveness-latency trade-off is vital for understanding the outcome: the best throughput is obtained for $r = 32$ and not $r = 128$. We note that our findings for MTPC-CP also hold for Llama and greedy speculative decoding, see Tables 10, 13 and 16 in the Appendix. In the last column, we also show the maximum attainable throughput ($\max_{\text{tok/s}}$), *i.e.*, we disable speculative decoding and accept all tokens. We see that we pay $\approx 90$ tok/s for $r = 32$, which is not a big price to pay for guaranteeing no loss in generation quality.

While MTPC-CP performs well for $n = 8$, the margin for further improving throughput is small. This is because for $n = 8$, we can at best achieve $\checkmark\mu_{\text{acc}} = 8$, and we have already achieved $\checkmark\mu_{\text{acc}} = 6.02$ and have hit diminishing returns. To obtain substantial boosts in throughput, we need to find another axis of expressivity to benefit from. We therefore use MTPC's more expressive circuits to extend our model to longer window sizes. As $r = 32$ worked best, we keep it fixed for the remaining experiments.

**RQ2**: Do we benefit from more expressive circuit architectures for longer MTP windows?

We now consider the more expressive MTPC-HMM and MTPC-BTREE, and show that they outperform MTPC-CP on $\checkmark\mu_{\text{acc}}$ for longer MTP windows, highlighting the importance of our extension to general PCs. We fix $r = 32$ and explore the different PC architectures for both $n = 8$ and the longer window, $n = 16$. Fig. 3 shows that both MTPC-HMM and MTPC-BTREE increase $\checkmark\mu_{\text{acc}}$. However, MTPC-HMM strikes an unfavourable balance in the expressiveness–latency trade-off: *Due to being AR,* MTPC-HMM *has the largest* $\circlearrowleft\mu_{lat}$ *(see Tables 8, 11, 14 and 17), and yields poor throughput as a result.*

Interestingly, for greedy decoding EvaByte-MTP achieves higher throughput than more expressive circuits (see Fig. 3, bottom right). We conjecture this is because EvaByte was pretrained with the independence assumption, so its representations are well-aligned with a factorised head such as

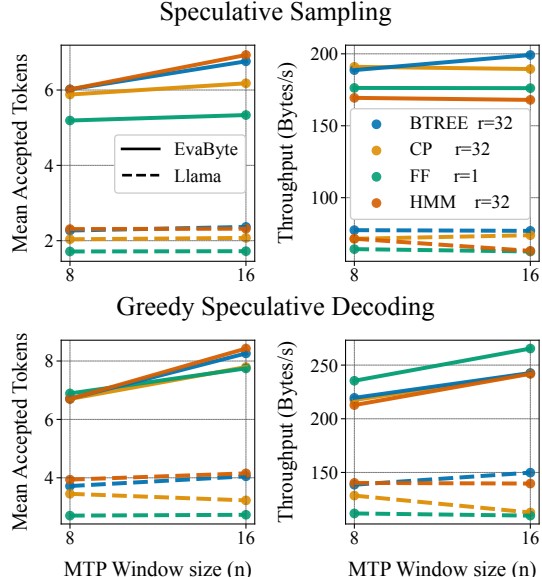

*Figure 3.* **Extending the MTP window of expressive PCs such as MTPC-HMM and MTPC-BTREE to** $n = 16$ **boosts their** ☑$\mu_{\mathrm{acc}}$ **in all settings**, as can be seen from the upward trend on the left subplots and in more detail in Fig. 6. Importantly, MTPC-BTREE improves $\mu_{\mathrm{tok/s}}$ for all but Llama with speculative sampling, highlighting its strong expressiveness–latency trade-off. In contrast, MTPC-HMM lags behind due to high latency from its AR nature.

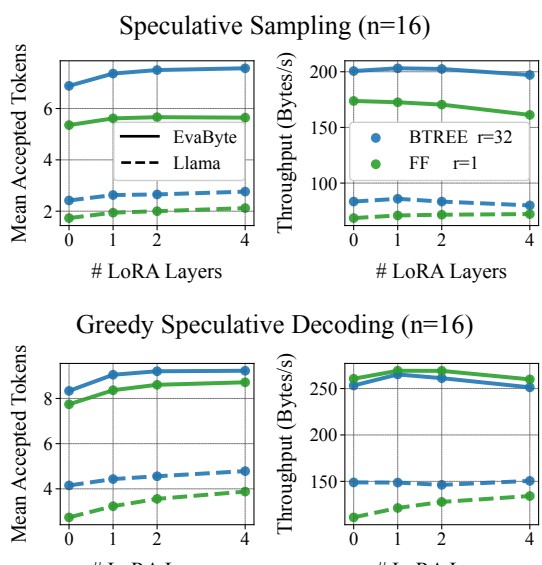

*Figure 4.* **We increase the expressiveness of the draft model by adding LoRA layers** (x-axis) for $n = 16$. As a result, the ☑$\mu_{\mathrm{acc}}$ (left subplots) increases substantially for both EvaByte and Llama for both speculative decoding settings. Due to the expressiveness–latency trade-off the optimal $\mu_{\mathrm{tok/s}}$ for EvaByte is obtained with 1 LoRA layer while Llama often benefits further from 2 or more.

MTPC-FF. This suggests that expressive circuits are not always necessary for greedy decoding, which requires only the correct argmax, but appear more beneficial for sampling, which requires approximating the full distribution.

While the gains already obtained by MTPC-BTREE are solid, fine-tuning the MTP head alone can only get us so far. This is because neither Llama nor EvaByte has been trained to produce representations that are good for predicting 16 tokens ahead, as we discuss next.

> TAKEAWAY 1: While increasing the mixture components $r$ in CP is initially beneficial, it soon hits diminishing returns. Increasing the MTP future token window size $n$ and adopting more expressive PC architectures unlocks further throughput gains. Moreover, while the HMM achieves the highest acceptance rates, it incurs high latency. Instead, *non-autoregressive* variants such as BTREE strike a better trade-off and should be preferred.

### 4.4. MTPCs with Adapters

**RQ3:** Can we further increase throughput by adapting the draft LLM using LoRA?

We now consider our final axis for increasing expressiveness: adding LoRA layers to the draft model. We note that we also fine-tune a model with $0$ LoRA layers for an additional epoch to highlight that our improvements are not due to longer training alone. We show that while we can improve throughput, we need to be strategic when choosing the number of LoRA layers, as the latency introduced can rapidly outweigh the expressiveness gained.

For example, if we train adapters for the last 16 (out of 32) layers of EvaByte, we can improve ☑$\mu_{\mathrm{acc}}$ by 37%, but we introduce a ⏱$\mu_{\mathrm{lat}}$ of $1.5\times$ the cost of a forward pass of the LLM.[10] As can be seen for $n = 8$ in Fig. 4, adding LoRA layers almost always improves throughput. An exception is EvaByte-MTP with $n = 8$ where the ☑$\mu_{\mathrm{acc}}$ cannot be increased by adding LoRA layers, see Fig. 7. This is likely because the backbone has been pretrained with the MTP loss, and its representations cannot be improved further. In contrast, EvaByte-MTP-BTree once again obtains larger increases in ☑$\mu_{\mathrm{acc}}$, especially for speculative sampling for $n = 16$ where by using 4 LoRA layers we obtain a boost of .69 in ☑$\mu_{\mathrm{acc}}$ over having no LoRA layers. However, because of the latency introduced, the best throughput is obtained with EvaByte-MTP-BTree with 1 LoRA layer, having a speed-up (⏩$\times_{\mathrm{STP}}$) of $\times 5.15$ over EvaByte-STP (see Table 9), while Llama benefits from 2 LoRA layers and obtains $\times 2.24$ speed-up over Llama-STP (see Table 15). Importantly, both obtain a $\times 1.17$ speed-up over the best FF MTP model, highlighting the importance of relaxing the independence assumptions to improve speculative sampling.

---

[10]We found that training more than 16 layers of EvaByte does not lead to improvements in acceptance rates.

TAKEAWAY 2: Fine-tuning a few layers of the draft model with LoRA increases the number of accepted tokens but also increases latency. The optimal trade-off can be device and backbone specific, but adding LoRAs is always beneficial compared to a fully shared LLM backbone, as long as we are retrofitting a model to an MTP window size it has not already been pretrained for.

## 5. Conclusions

We have identified key trade-offs between acceptance rates and latency within our framework, MTPC. We enhanced the *expressiveness* of MTP: we relaxed the independence assumption (Cai et al., 2024; Zheng et al., 2025) by introducing an explicit probabilistic model for inter-token dependencies, and generalised mixture-based methods (Basharin et al., 2025) into the PC framework, unlocking a better expressiveness-latency trade-off. We also showed how to further optimise the trade-off by modulating the number of layers shared between draft and verifier model backbones.

We showcased the throughput gains of MTPC LLMs *at scale* by retrofitting EvaByte (Zheng et al., 2025), a cutting-edge 6.5B byte-level LLM, and Llama 3.2 3B (Minixhofer et al., 2025), a byte-fied version of the widely used subword-level LLM, into our framework. Overall, our results show that throughput in MTP byte-level LLMs can be increased by $1.17\times$ over MTP with independence assumptions and $5.15\times/2.24\times$ over AR for EvaByte/Llama. Crucially, these speed-ups are obtained without sacrificing generation quality: we guarantee to match the target AR LLM exactly for greedy decoding (Stern et al., 2018) and in expectation for sampling (Leviathan et al., 2022).

In future work, our framework can be extended by integrating constraints during generation (Ahmed et al., 2025) or speculative decoding (Nakshatri et al., 2025) via methods such as Gelato (Zhang et al., 2023), Ctrl-G (Zhang et al., 2024a) and LTLA (Yidou-Weng et al., 2026). Unlike these methods, we would not need to train an auxiliary HMM in MTPC and could integrate constraints directly into our PC head. Moreover, we can further boost expressiveness by leveraging other PC architectures such as subtractive mixtures (Loconte et al., 2024; 2025b) and continuous latent variable circuits (Gala et al., 2024a;b), while reducing latency via recent advancements in scaling up PCs (Liu et al., 2024; Zhang et al., 2025).

## Limitations

**Batch Size**. We do not analyse the effect of using a batch size greater than one as we want to isolate our throughput gains to the expressiveness/latency trade-off. **Languages**. Byte-level LLMs are well-suited for multilingual generation

almost out-of-the-box as they can represent any word/script as a sequence of UTF-8 bytes. However, we only consider generation in English, as we found that EvaByte tended to switch between languages during generation and showed a large variance in throughput for different languages. While exploring the effect of multi-lingual generation is an interesting avenue for future work, we believe further pretraining of EvaByte or using other byte-level LLMs would be needed. **EOS Token**. In our experiments we restrict our models from producing the end-of-sequence token, as we want to avoid situations where some models obtain better throughput by generating shorter sequences. **MTPC on Subword-level LLMs**. While MTPC can be applied to subword-level LLMs with vocabulary sizes in the tens to hundreds of thousands, the amount of GPU memory needed during training to store the logits scales linearly in the vocabulary size as $s \times n \times v \times r$, where $s$ is the context length, $n$ the MTP window length, $v$ the vocabulary size and $r$ the number of mixture components. In the paper we use a context length $s = 8192$ for byte-level LLMs and we explored $r \in \{8, 16, 32, 64, 128\}$. However, for subword-level LLMs with a vocabulary size that is $100\times$ larger, we would hit a memory bottleneck with anything more than $r = 4$ on a GPU with the same memory (80 GB), as we show in Table 5 in Appendix G. We discuss approaches for mitigating this memory bottleneck in Appendix G. These are promising for extending MTPC to subword-level LLMs, but require additional ablations or non-trivial adaptations that were out of scope for this paper.

## Acknowledgements

We would like to thank the anonymous reviewers for their thorough feedback and suggestions. We also thank Lin Zheng, author of EvaByte, for help with the EvaByte codebase and for answering all of our questions on EvaByte. Moreover, we thank Benjamin Minixhofer for making their models available and for answering questions about the implementation. We would also like to thank the members of april lab and Ponti lab for useful feedback during early presentations of this work.

AG was funded by NatWest Group and ERC grant ("Numerical Analysis for Stable AI", 101198795). EvK was funded by the ELIAI project "Gradient-based Learning of Complex Latent Structures" and the NWO AiNed project "Human-Centric AI Agents with Common Sense" (NGF.1607.22.044). EP is supported by the ERC Starting Grant AToM-FM (101222956) "Adaptive Memory and Tokenization in Foundation Models for Efficient and Long-Horizon AI". AG and AV were funded by the "UNREAL: Unified Reasoning Layer for Trustworthy ML" project (EP/Y023838/1) selected by the ERC and funded by UKRI EPSRC.

## Impact statement

This paper presents work whose goal is to advance the field of Machine Learning. There are many potential societal consequences of our work, none which we feel must be specifically highlighted here.

## Reproducibility statement

To ensure reproducibility for our research, we have released the code and checkpoints for all model variants at github.com/april-tools/mtpc. In addition, we have provided the full details of sampling in circuits in Appendix B and of our algorithms for speculative decoding in Appendix F.

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

## A. Notation

We adapt notation and nomenclature from the tensor factorisation (Kolda & Bader, 2009) and circuit (Loconte et al., 2025a) literature.

We denote ordered sets of random variables with $\mathbf{X}$, $\mathbf{Y}$ and $\mathbf{Z}$, and we use $[n]$ to express the set $\{1, 2, \ldots, n\}$ with $n > 0$. The domain of a variable $X$ is denoted as $\mathsf{dom}(X)$, and we denoted as $\mathsf{dom}(\mathbf{X}) = \mathsf{dom}(X_1) \times \cdots \times \mathsf{dom}(X_n)$ the joint domain of variables $\mathbf{X} = \{X_i\}_{i=1}^n$. We denote scalars with lower-case letters (e.g., $a \in \mathbb{R}$), vectors with boldface lower-case letters (e.g., $\mathbf{a} \in \mathbb{R}^N$), matrices with boldface upper-case letters (excluding those used for variables, e.g., $\mathbf{A} \in \mathbb{R}^{M \times N}$), and tensors with boldface calligraphic letters (e.g., $\boldsymbol{\mathcal{A}} \in \mathbb{R}^{I_1 \times I_2 \times I_3}$). Moreover, we use subscripts to denote entries of tensors (e.g., $a_{ijk}$ is the $(i, j, k)$-th entry in $\boldsymbol{\mathcal{A}}$).

## B. Background on circuits

Circuits have a long history in theoretical computer science (Shpilka & Yehudayoff, 2010) and probabilistic reasoning (Darwiche, 2003; 2009). In their more modern definition and application to machine learning (Vergari et al., 2019b; Choi et al., 2020), circuits are introduced as structured computational graphs, simplified neural networks where one is allowed to use units from a restricted set of neurons (sum, product and input units) and whose connections need to abide certain *structural properties* to guarantee tractability (Choi et al., 2020; Vergari et al., 2021), as discussed next.

### B.1. Structural properties

Tractability is to be intended as the ability to exactly compute a given function (operation) over the circuit in time that is polynomial in its size, denoted as $|c|$ for a circuit $c$, and representing the number of edges between the computational units. For example, a circuit $c$ can exactly integrate *any subset of variables* in time $\mathcal{O}(|c|)$ if (i) its input functions can be integrated efficiently and (ii) it is *smooth* and *decomposable* (Darwiche & Marquis, 2002; Choi et al., 2020).

**Definition 1** (Smoothness and decomposability (Darwiche & Marquis, 2002; Choi et al., 2020))**.** A circuit is *smooth* if for every sum unit $n$, all its input units depend on the same variables, i.e., $\forall i, j \in \mathsf{in}(n)\colon \mathsf{sc}(i) = \mathsf{sc}(j)$. A circuit is *decomposable* if the distinct inputs of every product unit $n$ depend on disjoint sets of variables, i.e., $\forall i, j \in \mathsf{in}(n)\, i \neq j\colon \mathsf{sc}(i) \cap \mathsf{sc}(j) = \varnothing$.

Note that all the PC architectures we have discussed in this paper, FF, CP, HMM and BTree, are smooth and decomposable circuits by construction (Peharz et al., 2020c; Loconte et al., 2025a). The reader is encouraged to check this by themselves for the architectures in Fig. 2. Exactly integrating variables out is relevant to compute marginals such as the normalisation constant of the distribution encoded by the circuit (Eq. (3)). Note that in our implementation, circuits are normalised by design (Peharz et al., 2015), as we assume that input distributions are normalised categoricals and all sum units form a convex combination as their weights are parameterised with a softmax function (see Sec. 3.3).

More importantly for our MTPCs, we can draw samples efficiently from the distribution of a circuit that is both smooth and decomposable, as we discuss in the next sub-section.

### B.2. Sampling a circuit

A smooth and decomposable PC can use ancestral sampling to generate a complete sample for all $n$ tokens in a window. In a nutshell, we can iteratively sample each latent variable in the hierarchy encoded by the PC, and then sample the selected input distributions, in the same way one sample one (hierarchical) mixture model by first sampling one component and then drawing a sample from that component.

Operationally, Alg. 1 details the procedure. We have to sample one input branch for each sum unit we encounter when performing a backward traversal of the circuit computational graph (from the circuit output back to the input distributions). Such a branch is sampled proportionally to the sum unit weights $\omega_j$, which encode the mixture components (or equivalently the transition probabilities in an HMM). Then, when we traverse a product unit, we follow all its input branches. When we reach an input unit, we sample a token proportionally to the parameters $\phi_{ij}$ of the categorical distributions encoded in the unit. If the circuit is smooth and decomposable, by this process we are guaranteed to end up in a set of input units whose scope is the full set of tokens $\mathbf{X}$ and in which only one input unit is selected per token position $i$ (line 13 of Alg. 1). This procedure can be tensorised to efficiently generate a batch of samples in a single pass over the computational graph of the circuit (Vergari et al., 2019a; Peharz et al., 2020b;a; Loconte et al., 2025a; Liu et al., 2024).

---

**Algorithm 1** SAMPLE($c$)

---

**Input:** A smooth, decomposable and normalised PC $c$ encoding a joint distribution $q$ over the next $n$ tokens $\mathbf{X} = \{X_1, \ldots, X_n\}$ **Output:** a sample $\mathbf{x} \sim q(\mathbf{X})$.

1: $\mathbf{x} \leftarrow$ zeroes($n$) {init empty sample}
2: $c_n \leftarrow$ output($c$)
3: $\mathcal{N} \leftarrow$ queue($\{c_n\}$) {traverse the computational graph from outputs to inputs}
4: **while** $\mathcal{N}$ not empty **do**
5:     $c_n \leftarrow$ pop($\mathcal{N}$) {$c_n$ is a sum unit}
6:     **if** $c_n = \sum_{j=1}^r \omega_j c_j$ **then**
7:        $k \leftarrow$ sampleCategorical($\omega_1, \ldots, \omega_r$) {sample from a categorical with $r$ states}
8:        $\mathcal{N} \leftarrow$ push($\mathcal{N}, c_k$)
9:     **else if** $c_n = \prod_{j=1}^d c_j$ **then**
10:       **for** $k = 1 \ldots d$ **do**
11:          $\mathcal{N} \leftarrow$ push($\mathcal{N}, c_k$) {visit all inputs of product unit $c_n$}
12:       **end for**
13:     **else if** $c_n$ is an input unit over variable $X_i$ and parameters $\phi_i$ **then**
14:       $x_i \leftarrow$ sampleCategorical($\phi_i$) {sample from a categorical with $v = |\mathcal{V}|$ states}
15:     **end if**
16: **end while**
17: **return** $\mathbf{x}$

---

Lastly, we remark that this routine is potentially computationally more efficient than the one implemented in Basharin et al. (2025), as the latter is based on autoregressive inverse transform sampling (see Loconte et al. (2024) for a discussion) and requires sampling one token at a time.

## C. Complexity Analysis for Circuits in MTPC

We computed the theoretical number of FLOPs of the multi-token prediction linear head and the circuit feed-forward evaluation, by using calculate-flops.pytorch. We show the FLOPs needed to evaluate the circuit since it is required to compute the likelihoods in the speculative decoding algorithm. In Table 2, we report the number of parameters for combinations of circuits and the maximum number of draft tokens. We set the number of components of each sum to $r = 32$ in CP, HMM, and BTree. We also show the FLOPs required to do prefilling of a very small sequence of bytes (of length 64) with EvaByte for reference.

While using the MTPC-BTREE or MTPC-HMM increases the FLOPs to evaluate the circuit when compared to the fully factorised circuit, it is still orders of magnitudes smaller than the cost of evaluating the LLM trunk (i.e., a few thousands vs billions). Moreover, using the BTree or HMM substantially increases the parameters of the multi-token head linear projection, when compared to the fully factorised circuit. However, we note that it is still very close to the number of parameters required by MTPC-CP (i.e., 730 M vs 671 M in the case of 16 draft tokens). This is because the categorical input units account for the same vast majority of circuit parameters in all circuits except for the fully factorised one.

*Table 2.* FLOPs and parameter counts for circuit and MTP head configurations.

| LLM | FLOPs | # Tokens | Circuit | FLOPs | Parameters | MTP Head | FLOPs | Parameters |
|---|---|---|---|---|---|---|---|---|
| EvaByte | 828.95 G | | FF | 7 | 2,560 | Linear | 0.02 G | 10 M |
| | | 8 | CP | 65 | 81,952 | Linear | 0.67 G | 336 M |
| | | | HMM | 14.4 K | 89,120 | Linear | 0.73 G | 365 M |
| | | | BTree | 12.4 K | 88,096 | Linear | 0.72 G | 361 M |
| | | | FF | 15 | 5,120 | Linear | 0.04 G | 21 M |
| | | 16 | CP | 65 | 163,872 | Linear | 1.34 G | 671 M |
| | | | HMM | 30.8 K | 179,232 | Linear | 1.47 G | 734 M |
| | | | BTree | 28.7 K | 178,208 | Linear | 1.46 G | 730 M |

## D. Shallow Autoregressive Models for Byte-Level LLMs

We omit shallow autoregressive models such as Hydra (Ankner et al., 2024) and EAGLE (Li et al., 2024) from our setup as they do not fit our desiderata we mentioned in the introduction. At the same time, one may ask how well shallow autoregressive models perform for byte-level LLMs. While shallow autoregressive models are state-of-the-art for subword-level models, their superiority does not obviously transfer to byte-level models, and fairly comparing models from MTPC with them is non-trivial, as we discuss below.

### D.1. Key Differences between Byte-Level LLMs and Subword-Level LLMs

Byte-level models differ from subword-level models in two important ways: a) they need a larger MTP window size and b) they have a much smaller vocabulary size. These two differences alter the balance in parameter allocation and make it non-obvious that Hydra and EAGLE are state-of-the-art methods for byte-level LLMs (this remains to be seen).

**Window size differences.** To match subword-level throughput, byte-level models require MTP window sizes $n$ roughly 4-5 times longer, since 1 BPE token $\approx$ 4-5 bytes for English. Importantly, the memory required for the $d$-dimensional feature vectors for Hydra/Eagle scales as $\mathcal{O}(n \times s \times d)$, as the feature vector is adapted per token in the MTP window, in contrast to our MTPC feature embeddings which scale as $\mathcal{O}(s \times d)$. As such, when $n$ is larger, shallow autoregressive models are less appealing for MTP.

**Vocabulary differences.** Moreover, byte-level vocabularies are $100 - 400 \times$ smaller (e.g. $V_b$=320 vs $V_s$ in 32k - 128k), while $d$ is comparable (e.g. $d = 4096$ for EvaByte and $d = 3072$ for Llama). Using $d = 3200$ for simplicity, the logit-to-feature ratio for subword-level models is $V_s/D > 10$, while for byte models we have ($V_b/D < 1/10$). Thus, increasing expressivity in logit-space (as MTPC does) can be more efficient for byte-level models than increasing it in feature-space as done in Hydra/EAGLE.

### D.2. Comparing MTPC to Shallow Autoregressive Models is non-trivial

While we cannot make strong claims without experiments, our paper shows the expressiveness/latency trade-off is sensitive even to small changes, let alone differences of the magnitude mentioned above for the vocabulary size. This suggests that significant optimisation of the EAGLE/Hydra architectures may be needed for a fair comparison to MTPC on byte-level LLMs. We believe that optimising shallow autoregressive models for byte-level models is interesting, but we leave this for future work.

## E. Target Model Details

*Table 3.* Details of target models we retrofitted using MTPC.

|  | EvaByte | Llama3.2 3B Byte |
| --- | --- | --- |
| MTP Window | 8 | — |
| # Transformer Layers | 32 | 28 |
| # Attention Heads | 32 | 24 |
| Embedding Size | 4096 | 3072 |
| dtype | bfloat16 | float32 |
| Vocab Size | 320 | 268 |
| Number of Parameters | 6.5B | 3B |
| Max Context Window | 32768 | 131072 |

### E.1. EvaByte Details

In Table 4 we provide a copy of the benchmark results of fine-tuned version of EvaByte, EvaByte-SFT, which we retrofit in the paper. The model card can be found at https://hf.co/EvaByte/EvaByte-SFT, see also Zheng et al. (2025) for more details. The numbers in the table above were produced by Medusa-style (Cai et al., 2024) tree-based typical decoding. For all benchmarks apart from HumanEval, the results were produced via greedy decoding, *i.e.*, similar to (Stern

*Table 4.* Downstream benchmark performance of EvaByte-SFT, table taken verbatim from Zheng et al. (2025). Entries with † were computed by the EvaByte authors. All numbers were computed with Medusa-style tree-based greedy decoding using the multi-token head, apart from HumanEval*, which used typical sampling. The authors of EvaByte followed Tulu 3, and evaluated the Pass@10 rate for HumanEval with 20 samples at temperature 0.8.

| Model | BBH | GSM8k | IFEval | MATH | MMLU | HumanEval* | TruthQA |
|---|---|---|---|---|---|---|---|
| Gemma-2-9B-it | 20.0 | 79.7 | 69.9 | 29.8 | 69.1 | 71.7 | 61.4 |
| Ministral-8B-Instruct | 56.2 | 80.0 | 56.4 | 40.0 | 68.5 | 91.0 | 55.5 |
| Qwen-2.5-7B-Instruct | 25.3 | 83.8 | 74.7 | 69.9 | 76.6 | 93.1 | 63.1 |
| Llama-3.1-8B-Instruct | 69.7 | 83.4 | 80.6 | 42.5 | 71.3 | 86.3 | 55.1 |
| Tülu 3 8B | 66.0 | 87.6 | 82.4 | 43.7 | 68.2 | 83.9 | 55.0 |
| OLMo-7B-Instruct | 35.3 | 14.3 | 32.2 | 2.1 | 46.3 | 28.7† | 44.5 |
| OLMo-v1.7-7B-Instruct | 34.4 | 23.2 | 39.2 | 5.2 | 48.9 | 49.7† | 55.2 |
| OLMoE-1B-7B-0924-Instruct | 37.2 | 47.2 | 46.2 | 8.4 | 51.6 | 54.8 | 49.1 |
| MAP-Neo-7B-Instruct | 26.4 | 69.4 | 35.9 | 31.5 | 56.5 | 72.1† | 51.6 |
| OLMo-2-7B-SFT | 50.7 | 71.2 | 68.0 | 25.1 | 62.0 | 67.0† | 47.8 |
| OLMo-2-7B-1124-Instruct | 48.5 | 85.2 | 75.6 | 31.3 | 63.9 | 67.6† | 56.3 |
| EvaByte-SFT | 34.6 | 52.9 | 60.2 | 29.8 | 49.5 | 73.7 | 46.3 |

et al., 2018) with the exception of producing the last "free" token. As such, the produced tokens are equivalent to what EvaByte-STP would produce, and the authors found that the metrics were the same with EvaByte-STP up to some small rounding errors. For HumanEval the authors used tree-based typical decoding, which in this case does not maintain the quality of the EvaByte-STP model. The details above were shared with us by Lin Zheng, the author of EvaByte.

## F. Speculative Decoding

We give pseudocode for our self-speculative decoding algorithm below. The algorithm accepts between $0$ and $n$ tokens, but always generates between $1$ and $n$ tokens, where $n$ is the MTP window size. The algorithm is very similar to vanilla speculative decoding (Leviathan et al., 2022), but our algorithm includes a modification that reduces latency for the self-speculative scenario, and for this it needs to sacrifice the last "free" token typically obtained from the verifier. The gain in latency is possible because we can evaluate the shared LLM once per draft/verification cycle, while a naive implementation of Leviathan et al. (2022) for self-speculative decoding would need two, approximately halving the possible throughput.

In our self-speculative setup, the verifier and draft LLMs share some layers of the backbone. Importantly, the verifier is always computing LLM states ahead of the draft. As such, we can get away with a single forward pass through the shared LLM, similar to Medusa (Cai et al., 2024), by re-using the LLM backbone state computed by the verifier for the draft model. For this to work, we cannot accept a "last sample for free" from the verifier (lines 23-30) Alg. 3), as we would not have the backbone state for this new token and it is not worth paying an extra LLM evaluation for it. Therefore, in our algorithm we only sample the "free" token from the verifier in the rare case that no tokens are accepted. This is necessary because the model can get caught in successive no-accept states in the sampling case, or get stuck in an infinite loop if we use greedy decoding. If any tokens were accepted, we use the last state of the shared backbone computed during the verify phase to seed the draft phase. In what follows, if we have no LoRA layers, the algorithm is modified to have a single component: the shared encoder.

---

**Algorithm 2** SHAREDSTATESELFSPECULATIVEDECODING

---

**Architecture Components:**
    Three components: Shared Encoder ($S$), Verifier ($V$), Draft ($D$)
    Each with their own KV-cache
**Given:** A prompt of length $L$
**Initialisation:**
Prefill $V$ to $L - 1$, and $D$ and $S$ to $L$
Set $S$ and $D$ state

**Switch to draft/verify cycle:**
**while** true **do**
    **Draft stage:**
    **if** $S$ state is not set **then**
        Compute $S$ by conditioning on the additional token
    **end if**
    Use $S$ state to compute $D$ state
    Parameterize MTPC with $D$ state
    Draft $n$ tokens

    **Verify stage:**
    Compute $S$ state on $n + 1$ tokens (draft + predecessor)
    Compute $V$ state using $S$ state
    Obtain up to $n + 1$ tokens from speculative decoding
    **if** 0 tokens accepted **then**
        Keep "free" token sampled from last valid logits
        Unset $S$ state (stale)
    **else**
        Accept $n$ tokens (drop "free" token)
        Set $S$ state (hidden state for last accepted token)
    **end if**
**end while**

---

---

**Algorithm 3** SELFSPECULATIVEDECODING($\mathbf{x}_{\leq t}, f, h, c, g$)

---

**Input:** A prefix $\mathbf{x}_{\leq t}$ of length $t$, an LLM backbone $f \colon \mathcal{V}^* \to \mathbb{R}^d$, an LLM head $h \colon \mathbb{R}^d \to \boldsymbol{\Theta}$ parameterising a PC $c$ encoding a joint distribution $q$ over the next $n$ tokens, and an LLM head $g \colon \mathbb{R}^d \to \Delta^v$ computing the next token probabilities.

**Output:** A sentence $(\mathbf{x}_{\leq t} \| \mathbf{x}_{t+1:t+s}) \in \mathcal{V}^{t+s}$ where $1 \leq s \leq n+1$. Moreover, we have that $\mathbf{x}_{t+1:t+s} \sim p(x_{t+1}, \ldots, x_{t+s} \mid \mathbf{x}_{\leq t})$ as equivalently encoded by the autoregressive single token prediction model consisting of $f$ and $g$ only (Leviathan et al., 2022).

1: $\mathbf{e}_t \leftarrow f(\mathbf{x}_{\leq t})$ {Compute the last embedding}
2: $\boldsymbol{\theta} \leftarrow h(\mathbf{e}_t)$ {Compute the circuit parameters}
3:
4: Let $q(\mathbf{X}_{t+1:t+n} \mid \mathbf{x}_{\leq t}) = \frac{1}{Z_{\boldsymbol{\theta}}} c(\mathbf{X}_{t+1:t+n} \mid \boldsymbol{\theta})$
5: $\mathbf{x}_{t+1:t+n} \sim q(\mathbf{X}_{t+1:t+n} \mid \mathbf{x}_{\leq t})$ {Sample $n$ tokens from $c$ in time $\mathcal{O}(|c|)$}
6: $\mathbf{x} \leftarrow \mathbf{x}_{\leq t} \| \mathbf{x}_{t+1:t+n}$ {Concatenate the prefix with the $n$ tokens}
7:
8: Compute in parallel for $1 \leq i \leq n$: {Compute marginals in time $\mathcal{O}(|c|)$}
9: $\qquad q(\mathbf{x}_{t+1:t+i} \mid \mathbf{x}_{\leq t}) = \sum_{x_{t+i+1}, \ldots, x_{t+n} \in \mathcal{V}} q(\mathbf{x}_{t+1:t+n} \mid \mathbf{x}_{\leq t})$
10:
11: Compute in parallel for $1 \leq i \leq n+1$: {Compute target model conditionals}
12: $\qquad p(X_{t+i} \mid \mathbf{x}_{\leq t+i-1}) = g(\mathbf{e}_{t+i-1})$, where $\mathbf{e}_{t+i-1} = f(\mathbf{x}_{\leq t+i-1})$
13:
14: $s \leftarrow 0$ {Determine the number of accepted tokens $s$, $0 \leq s \leq n$}
15: **while** $s < n$ **do**
16: $\quad \alpha \sim \mathcal{U}(0, 1)$
17: $\quad$ **if** $s > 0$ **then**
18: $\qquad q(x_{t+s+1} \mid \mathbf{x}_{\leq t+s}) \leftarrow q(\mathbf{x}_{t+1:t+s+1} \mid \mathbf{x}_{\leq t})/q(\mathbf{x}_{t+1:t+s} \mid \mathbf{x}_{\leq t})$
19: $\quad$ **end if**
20: $\quad$ **if** $\alpha > p(x_{t+s+1} \mid \mathbf{x}_{t+s})/q(x_{t+s+1} \mid \mathbf{x}_{t+s})$ **then**
21: $\qquad$ **exit loop**
22: $\quad$ **end if**
23: $\quad s \leftarrow s + 1$
24: **end while**{Sample one last token from the autoregressive LLM model}
25: **if** $s < n$ **then**
26: $\quad$ Let $s(X_{t+s+1}) = q(X_{t+s+1} \mid \mathbf{x}_{\leq t+s})$ {Adjust the distribution first, if we accept fewer tokens}
27: $\quad$ Let $m(X_{t+s+1}) = \max(0, p(X_{t+s+1} \mid \mathbf{x}_{\leq t+s}) - s(X_{t+s+1}))$
28: $\quad r(X_{t+s+1} \mid \mathbf{x}_{t+s}) = m(X_{t+s+1})/Z$, with $Z = \sum_{x' \in \mathcal{V}} m(x')$
29: $\quad x_{t+s+1} \sim r(X_{t+s+1} \mid \mathbf{x}_{\leq t+s})$
30: **else**
31: $\quad x_{t+s+1} \sim p(X_{t+s+1} \mid \mathbf{x}_{\leq t+s})$
32: **end if**
33: **return** $\mathbf{x}_{\leq t+s} \| x_{t+s+1}$

---

## F.1. GPU Memory Usage For Speculative Decoding

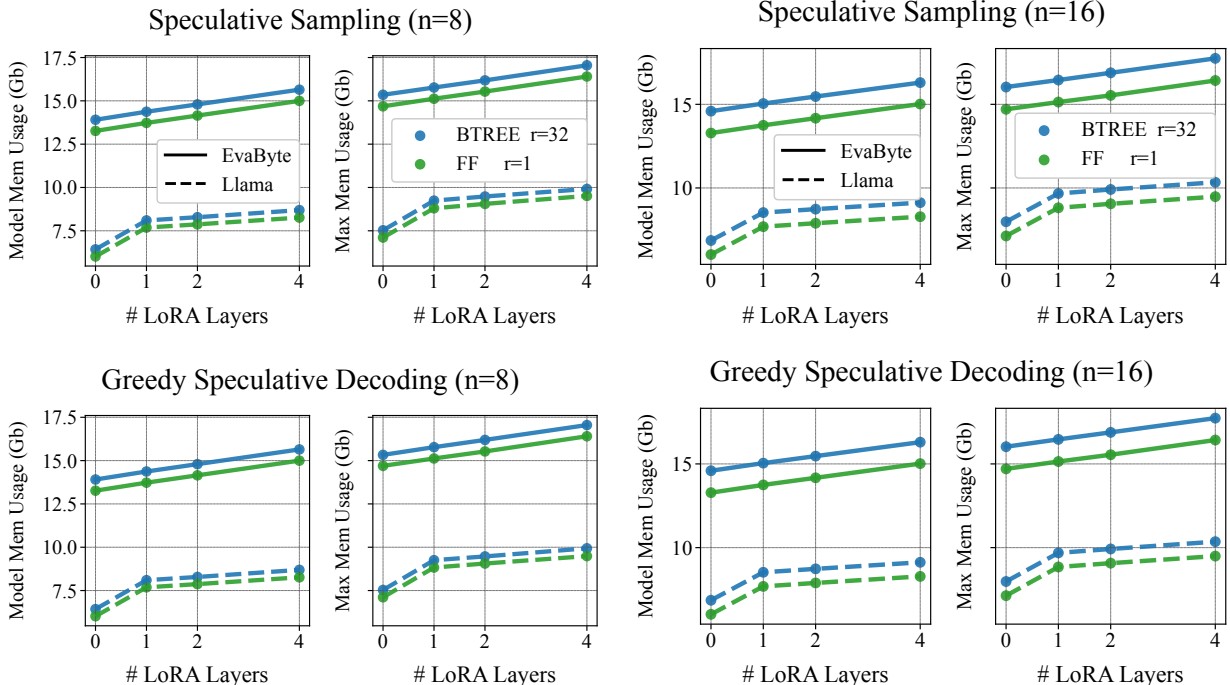

*Figure 5.* GPU memory usage as a function of the number of LoRA layers (x-axis) during model loading (left subplot) and maximum memory usage during inference (right subplot). As can be seen, MTPC-BTREE has similar memory requirements to MTPC-FF and scales graciously with the MTP window size, $n$. Moreover, retrofitting LoRA on the last 1-4 layers barely changes the footprint: we pay a memory overhead of approximately k transformer layers when using k LoRA layers. Measurements taken on the L40S GPUs during the speculative decoding experiments with KV cache enabled, BF16 computations, and 1,024 generated tokens.

## G. Memory Bottleneck Due to Logits

In this section we elaborate on the memory bottleneck that arises in MTP, especially when we use a model that has mixture coefficients, *e.g.* MTPC-CP, and propose ways to alleviate this bottleneck. As we mentioned in Sec. 5, the GPU memory required during training to store the logits scales linearly in the vocabulary size as $s \times n \times v \times r$, where $s$ is the context length, $n$ the MTP window length, $v$ the vocabulary size and $r$ the number of mixture components. As can be seen in Table 5, the size of the logits quickly becomes too large to fit in GPU memory if we scale the vocabulary size. Therefore, although we can scale MTPC to subword-level models with vocabulary sizes of 32k+, we are forced to use a much smaller number of mixture coefficients, making the approach less attractive. However, there are several strategies to keep the parameter count under control despite the increase in vocabulary size from 320 to 32k+ which can be explored in follow-up work.

**Subsampling MTP windows.** One pragmatic way of reducing the memory requirements due to logits during training is to avoid realising all of them. For example, we could subsample the token windows used for training in the MTP window.

**Unembedding parameter sharing.** We could share parameters across token positions and the mixture components and fine-tune low-rank adaptors (Hu et al., 2022) to allow some flexibility while keeping the parameter count under control. For example, we could parametrise the unembedding matrix for token $i$ and mixture component $j$ as $\mathbf{U}_{i,j} = \mathbf{W}_i + \mathbf{L}_{i,j}$, where $\mathbf{W}_i \in \mathbb{R}^{v \times d}$ is the pretrained unembedding matrix for token $i$ and $\mathbf{L}_{i,j} = \mathbf{A}_{i,j}\mathbf{B}_{i,j}$ is a rank-$k$ LoRA adaptor where $\mathbf{A}_{i,j} \in \mathbb{R}^{v \times k}, \mathbf{B}_{i,j} \in \mathbb{R}^{k \times d}$. For EvaByte, the pretrained unembedding matrices are available and we smoothly resume training from them by setting all entries of $\mathbf{B}_{i,j}$ to zero, while for other models we would need to randomly initialise the unembedding matrices.

**Low-rank unembedding matrices.** Another option is to explore whether we can lower the rank of the unembedding matrices (*i.e.* make $d$ smaller) without losing performance. While it is known that a low-rank softmax layer is less

*Table 5.* Memory footprint of the logits for MTPC-CP as a function of the vocabulary size $v$, the MTP window size $n$, and the number of mixture coefficients $r$ for a sequence length of 8192. While the logits are manageable for a byte-level model with a vocabulary of size $v = 320$ even for large $r$, memory scales linearly in $v$, so for subword-level models with large vocabularies the requirement quickly exceeds that available on our GPU (80 GB).

| | | Size (float 32 precision) | |
| --- | --- | --- | --- |
| $v$ | $r$ | $n = 8$ | $n = 16$ |
| | 1 | 80.0 MB | 160.0 MB |
| | 4 | 320.0 MB | 640.0 MB |
| 320 | 8 | 640.0 MB | 1.2 GB |
| | 16 | 1.2 GB | 2.5 GB |
| | 32 | 2.5 GB | 5.0 GB |
| | 1 | 7.8 GB | 15.6 GB |
| | 4 | 31.2 GB | 62.5 GB |
| 32,000 | 8 | 62.5 GB | 125.0 GB |
| | 16 | 125.0 GB | 250.0 GB |
| | 32 | 250.0 GB | 500.0 GB |
| | 1 | 31.2 GB | 62.5 GB |
| | 4 | 125.0 GB | 250.0 GB |
| 128,000 | 8 | 250.0 GB | 500.0 GB |
| | 16 | 500.0 GB | 1000.0 GB |
| | 32 | 1.0 TB | 2.0 TB |

expressive (Yang et al., 2018), the solution proposed in Yang et al. (2018, Section 2.4) is to increase expressivity by using a mixture of softmaxes. Since we are already employing a mixture of softmaxes (albeit across multiple tokens), we believe it may be possible to shrink $d$ without sacrificing too much performance.

**Hierarchical partitioning and tensor factorisations.** Lastly, we can explore ways of partitioning large vocabularies, *e.g.* hierarchical softmax (Mnih & Hinton, 2008), which introduces a non-linearity or other multi-linear tensor factorisations. For example, we could partition the vocabulary for each token into $m$ categories and introduce a latent variable to switch between the categories for each token position. This can be beneficial because within each category we can limit the active vocabulary size significantly, and can therefore use an unembedding matrix which is lower rank, as proposed previously.

## H. Hidden Markov Models Setup

In our experiments we use **contextual**, **inhomogeneous** hidden Markov models (HMMs) with **identity initialisation** (see Table 6). We chose the above after preliminary experiments where we assessed the following configuration choices for training our HMMs.

**Parameterisation.** We can parameterise HMMs to either be contextual, i.e. we can make the transition probabilities depend on the input, or we can make the transition probabilities be independent of the input (non-contextual).

**Transition type.** The transition matrix can be the same at each time step (homogeneous) or it can be different (inhomogeneous). The former would correspond to additional parameter sharing across sum layers in the circuit representation. We note that inhomogeneous HMMs subsume homogeneous HMMs. This is because inhomogeneous HMMs could in theory learn parameters that do not vary from timestep to timestep, thus becoming equivalent to a homogeneous HMM.

**Initialisation.** A crucial setup is to initialise the HMM with transition matrices that are identity matrices, which make the HMM equivalent to CP at the beginning of training. We achieve this by adding a bias term to allow the HMM model to be initialised to identity matrices. This setting in combination with extending to larger token windows, i.e. $n = 16$ lead to a scenario where HMMs outperform CP. The other alternative is to initialise the transition matrices uniformly at random (before softmax), but this complicates learning and yields performance that is lower than CP models.

*Table 6.* Most Successful HMM Configuration

| Parameterisation | | Transition Type | | Initialisation | |
|---|---|---|---|---|---|
| Contextual | Non-Contextual | Homogeneous | Inhomogeneous | Identity Init. | Uniform Init. |
| ✓ | ✗ | ✗ | ✓ | ✓ | ✗ |

# I. Further Experimental Details

To make the comparison between methods fair, we: a) constrained the models to not produce end-of-sequence symbols during generation, as the latency of retrieving KV cache items from memory increases with sequence length (Nawrot et al., 2024) and b) we filtered the validation set of the models to only include examples with both prompts and responses in English, as acceptance rates may vary dramatically based on the language chosen for the response.

We compute throughput by generating answers to 250 prompts and report the mean and std of 3 runs with different prompts.

# J. Alternative Losses

In early versions of this work we also experimented using a Kullback-Leibler divergence (KL) loss as recommended by Basharin et al. (2025). However, we found that training with the KL loss doubled the training time while requiring a lot more memory, and the benefits in the number of accepted tokens did not outweigh the additional complexity. For completeness we include the loss below. **KL Loss** $\mathcal{L}$

$$\mathcal{L} = \sum_{j=1}^{n} \mathcal{L}_j \gamma^{j-1}, \quad \mathcal{L}_j = \sum_{i=1}^{N} \sum_{t=1}^{L} \frac{f_{\text{KL}}\left(p_\theta(x_{t+j}^{(i)} \mid \mathbf{x}_{<t+j}^{(i)}) \,\middle\|\, q_{\theta'}(x_{t+j}^{(i)} \mid \mathbf{x}_{<t+j}^{(i)})\right)}{N\text{valid}(i,j)} \tag{6}$$

In the above we condition both the draft model, $q_{\theta'}$, and the target model, $p_\theta$, on the gold data. The above is equivalent to the KL term from the word-level distillation loss in (Kim & Rush, 2016).[11]

---

[11]While performing sequence-level distillation, *i.e.*, conditioning on data sampled from the teacher model may improve distillation (Kim & Rush, 2016), we did not explore this.

# K. Results on L40S GPU

## K.1. EvaByte 6.5B

### K.1.1. SPECULATIVE SAMPLING

*Table 7*

|  |  | ⏱ $\mu_{\text{lat}}$ ↓ | ☑ $\mu_{\text{acc}}$ ↑ | $\mu_{\text{tok/s}}$ ↑ | $\max_{\text{tok/s}}$ |
|---|---|---|---|---|---|
| FF | 1 | **0.0299**±0.0003 | 5.19±0.05 | 176.3±0.9 | 297.55 |
|  | 8 | 0.0314±0.0005 | 5.67±0.05 | 184.2±3.9 | 297.92 |
|  | 16 | 0.0323±0.0020 | 5.79±0.08 | 183.4±13.3 | 290.71 |
| CP | 32 | 0.0314±0.0003 | 5.88±0.02 | **191.0**±1.6 | 287.88 |
|  | 64 | 0.0327±0.0015 | 5.96±0.04 | 185.9±7.7 | 281.10 |
|  | 128 | 0.0337±0.0001 | **6.02**±0.03 | 182.3±1.1 | 264.80 |

*Table 8*

|  |  |  | ⏱ $\mu_{\text{lat}}$ ↓ | ☑ $\mu_{\text{acc}}$ ↑ | $\mu_{\text{tok/s}}$ ↑ | ⏭ ×STP |
|---|---|---|---|---|---|---|
| 1 | 1 | STP | **0.0254** | — | 39.5 | 1.00 |
| 8 | 1 | FF | 0.0299±0.0003 | 5.19±0.05 | 176.3±0.9 | 4.47 |
|  | 32 | HMM | 0.0362±0.0006 | **6.02**±0.02 | 169.4±2.6 | 4.29 |
|  |  | BTree | 0.0326±0.0008 | **6.02**±0.05 | 188.7±6.1 | 4.78 |
|  |  | CP | 0.0314±0.0003 | 5.88±0.02 | **191.0**±1.6 | **4.84** |
| 16 | 1 | FF | 0.0308±0.0003 | 5.34±0.03 | 176.1±2.3 | 4.46 |
|  | 32 | HMM | 0.0421±0.0022 | **6.93**±0.06 | 168.0±9.4 | 4.26 |
|  |  | CP | 0.0333±0.0009 | 6.18±0.02 | 189.4±5.4 | 4.80 |
|  |  | BTree | 0.0347±0.0000 | 6.76±0.07 | **199.2**±2.2 | **5.05** |

*Table 9*

|  |  |  | ⏱ $\mu_{\text{lat}}$ ↓ | ☑ $\mu_{\text{acc}}$ ↑ | $\mu_{\text{tok/s}}$ ↑ | ⏭ ×STP |
|---|---|---|---|---|---|---|
| 1 | STP | — | **0.0254** | — | 39.5 | 1.00 |
| 8 | FF | 0 | 0.0308±0.0004 | 5.11±0.03 | 168.9±2.9 | 4.28 |
|  |  | 1 | 0.0325±0.0011 | 5.11±0.07 | 160.0±7.8 | 4.06 |
|  |  | 2 | 0.0332±0.0009 | 5.12±0.07 | 157.1±6.3 | 3.98 |
|  |  | 4 | 0.0348±0.0004 | 5.12±0.04 | 149.5±3.1 | 3.79 |
|  | BTree | 0 | 0.0332±0.0006 | 6.05±0.05 | **186.1**±1.8 | **4.72** |
|  |  | 1 | 0.0348±0.0010 | 6.16±0.06 | 180.9±6.9 | 4.58 |
|  |  | 2 | 0.0354±0.0014 | 6.18±0.03 | 178.2±7.9 | 4.52 |
|  |  | 4 | 0.0373±0.0017 | **6.22**±0.01 | 170.9±7.5 | 4.33 |
| 16 | FF | 0 | 0.0313±0.0012 | 5.36±0.07 | 173.8±7.9 | 4.40 |
|  |  | 1 | 0.0331±0.0010 | 5.62±0.08 | 172.6±7.1 | 4.37 |
|  |  | 2 | 0.0339±0.0010 | 5.66±0.15 | 170.5±8.9 | 4.32 |
|  |  | 4 | 0.0357±0.0012 | 5.64±0.11 | 161.2±8.7 | 4.09 |
|  | BTree | 0 | 0.0350±0.0002 | 6.88±0.10 | 200.7±2.6 | 5.09 |
|  |  | 1 | 0.0371±0.0008 | 7.36±0.03 | **203.1**±5.0 | **5.15** |
|  |  | 2 | 0.0379±0.0013 | 7.50±0.03 | 202.5±7.0 | 5.13 |
|  |  | 4 | 0.0393±0.0006 | **7.57**±0.13 | 197.1±6.3 | 5.00 |

### K.1.2. GREEDY SPECULATIVE DECODING

*Table 10*

|  |  | $\bigcirc\mu_{\text{lat}}\downarrow$ | $\checkmark\mu_{\text{acc}}\uparrow$ | $\mu_{\text{tok/s}}\uparrow$ | $\max_{\text{tok/s}}$ |
|---|---|---|---|---|---|
| FF | 1 | **0.0300**±0.0006 | **6.89**±0.01 | **235.4**±5.4 | 299.64 |
|  | 8 | 0.0310±0.0002 | 6.68±0.00 | 221.5±1.3 | 295.79 |
|  | 16 | 0.0312±0.0000 | 6.70±0.02 | 221.3±0.5 | 294.16 |
| CP | 32 | 0.0317±0.0002 | 6.70±0.01 | 217.4±1.6 | 288.68 |
|  | 64 | 0.0319±0.0002 | 6.69±0.01 | 215.9±1.1 | 282.55 |
|  | 128 | 0.0340±0.0003 | 6.68±0.02 | 202.3±2.1 | 264.70 |

*Table 11*

|  |  |  | $\bigcirc\mu_{\text{lat}}\downarrow$ | $\checkmark\mu_{\text{acc}}\uparrow$ | $\mu_{\text{tok/s}}\uparrow$ | $\blacktriangleright\blacktriangleright\times_{\text{STP}}$ |
|---|---|---|---|---|---|---|
| 1 | 1 | STP | **0.0251** | — | 39.9 | 1.00 |
|  | 1 | FF | 0.0300±0.0006 | **6.89**±0.01 | **235.4**±5.4 | 5.90 |
| 8 | 32 | HMM | 0.0324±0.0001 | 6.70±0.03 | 212.8±1.5 | 5.33 |
|  |  | CP | 0.0317±0.0002 | 6.70±0.01 | 217.4±1.6 | 5.45 |
|  |  | BTree | 0.0315±0.0004 | 6.71±0.03 | 219.5±3.6 | 5.50 |
|  | 1 | FF | 0.0299±0.0002 | 7.75±0.02 | **265.4**±1.8 | **6.65** |
| 16 | 32 | HMM | 0.0360±0.0009 | **8.43**±0.02 | 241.8±6.0 | 6.06 |
|  |  | CP | 0.0333±0.0002 | 7.79±0.05 | 242.2±1.8 | 6.07 |
|  |  | BTree | 0.0353±0.0020 | 8.26±0.08 | 242.6±12.4 | 6.08 |

*Table 12*

|  |  |  | $\bigcirc\mu_{\text{lat}}\downarrow$ | $\checkmark\mu_{\text{acc}}\uparrow$ | $\mu_{\text{tok/s}}\uparrow$ | $\blacktriangleright\blacktriangleright\times_{\text{STP}}$ |
|---|---|---|---|---|---|---|
| 1 | STP | — | **0.0251** | — | 39.9 | 1.00 |
|  |  | 0 | 0.0297±0.0003 | 6.89±0.01 | 237.3±2.2 | 5.95 |
|  | FF | 1 | 0.0311±0.0003 | 6.89±0.01 | **227.2**±2.7 | **5.69** |
|  |  | 2 | 0.0319±0.0002 | **6.90**±0.02 | 221.2±1.7 | 5.54 |
|  |  | 4 | 0.0336±0.0006 | 6.88±0.01 | 209.6±4.1 | 5.25 |
| 8 |  | 0 | 0.0316±0.0004 | 6.72±0.02 | 218.6±2.6 | 5.48 |
|  | BTree | 1 | 0.0333±0.0006 | 6.73±0.01 | 208.3±3.6 | 5.22 |
|  |  | 2 | 0.0337±0.0002 | 6.73±0.01 | 205.6±1.2 | 5.15 |
|  |  | 4 | 0.0356±0.0009 | 6.72±0.03 | 194.5±4.0 | 4.87 |
|  |  | 0 | 0.0305±0.0005 | 7.74±0.03 | 260.6±4.5 | 6.53 |
|  | FF | 1 | 0.0319±0.0004 | 8.36±0.04 | **269.2**±3.4 | **6.74** |
|  |  | 2 | 0.0329±0.0002 | 8.60±0.02 | 269.1±1.9 | 6.74 |
|  |  | 4 | 0.0346±0.0002 | 8.72±0.02 | 259.9±1.1 | 6.51 |
| 16 |  | 0 | 0.0341±0.0003 | 8.33±0.10 | 253.2±5.0 | 6.35 |
|  | BTree | 1 | 0.0354±0.0001 | 9.05±0.06 | 265.2±2.3 | 6.65 |
|  |  | 2 | 0.0365±0.0002 | 9.20±0.09 | 261.2±3.6 | 6.54 |
|  |  | 4 | 0.0381±0.0001 | **9.23**±0.07 | 251.3±2.5 | 6.30 |

## K.2. Llama 3.2 3B

### K.2.1. SPECULATIVE SAMPLING

*Table 13*

|  |  | $\bigcirc \mu_{\text{lat}} \downarrow$ | $\checkmark \mu_{\text{acc}} \uparrow$ | $\mu_{\text{tok/s}} \uparrow$ | $\max_{\text{tok/s}}$ |
|---|---|---|---|---|---|
| FF | 1 | $0.0291_{\pm 0.0031}$ | $1.71_{\pm 0.01}$ | $64.2_{\pm 6.4}$ | 342.39 |
|  | 8 | $0.0298_{\pm 0.0031}$ | $1.92_{\pm 0.04}$ | $70.2_{\pm 5.9}$ | 355.81 |
| CP | 16 | $0.0301_{\pm 0.0036}$ | $1.98_{\pm 0.02}$ | $\mathbf{71.7}_{\pm 8.0}$ | 357.04 |
|  | 32 | $\mathbf{0.0311}_{\pm 0.0032}$ | $\mathbf{2.04}_{\pm 0.04}$ | $71.4_{\pm 8.2}$ | 357.82 |

*Table 14*

|  |  |  | $\bigcirc \mu_{\text{lat}} \downarrow$ | $\checkmark \mu_{\text{acc}} \uparrow$ | $\mu_{\text{tok/s}} \uparrow$ | $\blacktriangleright\blacktriangleright\times_{\text{STP}}$ |
|---|---|---|---|---|---|---|
| 1 | 1 | STP | $\mathbf{0.0256}$ | — | 39.2 | 1.00 |
|  | 1 | FF | $0.0291_{\pm 0.0031}$ | $1.71_{\pm 0.01}$ | $64.2_{\pm 6.4}$ | 1.64 |
| 8 |  | CP | $0.0311_{\pm 0.0032}$ | $2.04_{\pm 0.04}$ | $71.4_{\pm 8.2}$ | 1.82 |
|  | 32 | HMM | $0.0351_{\pm 0.0030}$ | $\mathbf{2.31}_{\pm 0.05}$ | $71.5_{\pm 6.7}$ | 1.82 |
|  |  | BTree | $0.0318_{\pm 0.0040}$ | $2.27_{\pm 0.04}$ | $\mathbf{77.4}_{\pm 8.7}$ | $\mathbf{1.98}$ |
|  | 1 | FF | $0.0299_{\pm 0.0023}$ | $1.72_{\pm 0.03}$ | $62.6_{\pm 4.3}$ | 1.60 |
| 16 |  | HMM | $0.0400_{\pm 0.0045}$ | $2.32_{\pm 0.03}$ | $63.0_{\pm 6.3}$ | 1.61 |
|  | 32 | CP | $0.0303_{\pm 0.0028}$ | $2.07_{\pm 0.01}$ | $73.9_{\pm 6.8}$ | 1.89 |
|  |  | BTree | $0.0333_{\pm 0.0037}$ | $\mathbf{2.37}_{\pm 0.05}$ | $\mathbf{76.9}_{\pm 6.9}$ | $\mathbf{1.96}$ |

*Table 15*

|  |  |  | $\bigcirc \mu_{\text{lat}} \downarrow$ | $\checkmark \mu_{\text{acc}} \uparrow$ | $\mu_{\text{tok/s}} \uparrow$ | $\blacktriangleright\blacktriangleright\times_{\text{STP}}$ |
|---|---|---|---|---|---|---|
| 1 | STP |  | — | $\mathbf{0.0256}$ | — | 39.2 | 1.00 |
|  |  | 0 | $0.0269_{\pm 0.0001}$ | $1.73_{\pm 0.02}$ | $69.4_{\pm 0.9}$ | 1.77 |
|  | FF | 1 | $0.0323_{\pm 0.0041}$ | $1.95_{\pm 0.06}$ | $67.0_{\pm 9.4}$ | 1.71 |
|  |  | 2 | $0.0308_{\pm 0.0003}$ | $2.02_{\pm 0.08}$ | $72.1_{\pm 3.1}$ | 1.84 |
| 8 |  | 4 | $0.0318_{\pm 0.0001}$ | $2.18_{\pm 0.04}$ | $75.2_{\pm 1.0}$ | 1.92 |
|  |  | 0 | $0.0301_{\pm 0.0004}$ | $2.31_{\pm 0.02}$ | $82.9_{\pm 0.6}$ | 2.12 |
|  | BTree | 1 | $0.0334_{\pm 0.0021}$ | $2.54_{\pm 0.02}$ | $82.0_{\pm 5.8}$ | 2.09 |
|  |  | 2 | $0.0332_{\pm 0.0011}$ | $2.71_{\pm 0.04}$ | $\mathbf{87.7}_{\pm 1.9}$ | $\mathbf{2.24}$ |
|  |  | 4 | $0.0362_{\pm 0.0042}$ | $\mathbf{2.82}_{\pm 0.08}$ | $84.1_{\pm 10.1}$ | 2.15 |
|  |  | 0 | $0.0273_{\pm 0.0002}$ | $1.73_{\pm 0.04}$ | $68.6_{\pm 0.6}$ | 1.75 |
|  | FF | 1 | $0.0300_{\pm 0.0004}$ | $1.94_{\pm 0.03}$ | $70.9_{\pm 1.3}$ | 1.81 |
|  |  | 2 | $0.0308_{\pm 0.0002}$ | $2.00_{\pm 0.02}$ | $71.6_{\pm 0.1}$ | 1.83 |
| 16 |  | 4 | $0.0323_{\pm 0.0005}$ | $2.12_{\pm 0.05}$ | $72.2_{\pm 1.2}$ | 1.84 |
|  |  | 0 | $0.0312_{\pm 0.0004}$ | $2.42_{\pm 0.03}$ | $83.4_{\pm 1.4}$ | 2.13 |
|  | BTree | 1 | $0.0329_{\pm 0.0006}$ | $2.63_{\pm 0.07}$ | $\mathbf{85.9}_{\pm 3.3}$ | $\mathbf{2.19}$ |
|  |  | 2 | $0.0343_{\pm 0.0002}$ | $2.65_{\pm 0.03}$ | $83.4_{\pm 1.2}$ | 2.13 |
|  |  | 4 | $0.0376_{\pm 0.0041}$ | $\mathbf{2.76}_{\pm 0.02}$ | $80.0_{\pm 8.3}$ | 2.04 |

### K.2.2. GREEDY SPECULATIVE DECODING

*Table 16*

|  |  | $\bigcirc \mu_{\text{lat}} \downarrow$ | $\blacksquare \mu_{\text{acc}} \uparrow$ | $\mu_{\text{tok/s}} \uparrow$ | $\max_{\text{tok/s}}$ |
|---|---|---|---|---|---|
| FF | 1 | **0.0248**±0.0010 | 2.70±0.00 | 111.8±4.3 | 330.31 |
|  | 8 | 0.0264±0.0003 | 3.22±0.03 | 128.2±1.1 | 310.61 |
| CP | 16 | 0.0286±0.0020 | 3.39±0.05 | 125.2±7.9 | 315.91 |
|  | 32 | 0.0285±0.0026 | **3.45**±0.04 | **128.4**±12.2 | 282.88 |

*Table 17*

|  |  |  | $\bigcirc \mu_{\text{lat}} \downarrow$ | $\blacksquare \mu_{\text{acc}} \uparrow$ | $\mu_{\text{tok/s}} \uparrow$ | $\blacktriangleright\!\blacktriangleright \times_{\text{STP}}$ |
|---|---|---|---|---|---|---|
| 1 | 1 | STP | 0.0255 | — | 39.4 | 1.00 |
|  | 1 | FF | **0.0248**±0.0010 | 2.70±0.00 | 111.8±4.3 | 2.84 |
| 8 | | CP | 0.0285±0.0026 | 3.45±0.04 | 128.4±12.2 | 3.26 |
| | 32 | BTree | 0.0283±0.0022 | 3.72±0.10 | 138.6±7.7 | 3.52 |
|  | | HMM | 0.0295±0.0023 | **3.94**±0.06 | **140.3**±11.5 | **3.56** |
|  | 1 | FF | 0.0257±0.0027 | 2.73±0.01 | 109.8±11.2 | 2.79 |
| 16 | | CP | 0.0304±0.0025 | 3.23±0.06 | 112.7±9.9 | 2.86 |
| | 32 | HMM | 0.0312±0.0023 | **4.15**±0.07 | 139.6±8.0 | 3.55 |
|  | | BTree | 0.0285±0.0001 | 4.05±0.06 | **149.8**±1.9 | **3.81** |

*Table 18*

|  |  |  | $\bigcirc \mu_{\text{lat}} \downarrow$ | $\blacksquare \mu_{\text{acc}} \uparrow$ | $\mu_{\text{tok/s}} \uparrow$ | $\blacktriangleright\!\blacktriangleright \times_{\text{STP}}$ |
|---|---|---|---|---|---|---|
| 1 | STP | — | 0.0255 | — | 39.4 | 1.00 |
|  |  | 0 | **0.0252**±0.0014 | 2.71±0.00 | 110.8±6.2 | 2.81 |
|  | FF | 1 | 0.0266±0.0003 | 3.27±0.03 | 126.8±0.6 | 3.22 |
|  |  | 2 | 0.0288±0.0018 | 3.57±0.04 | 128.4±8.0 | 3.26 |
|  |  | 4 | 0.0300±0.0003 | 3.89±0.06 | 134.8±3.2 | 3.42 |
| 8 |  | 0 | 0.0287±0.0017 | 3.86±0.07 | 141.7±8.8 | 3.60 |
|  | BTree | 1 | 0.0291±0.0005 | 4.12±0.05 | 148.7±0.8 | 3.78 |
|  |  | 2 | 0.0297±0.0002 | 4.29±0.03 | **151.7**±1.8 | **3.85** |
|  |  | 4 | 0.0320±0.0019 | **4.48**±0.05 | 147.0±9.1 | 3.73 |
|  |  | 0 | 0.0253±0.0016 | 2.74±0.01 | 111.2±7.0 | 2.82 |
|  | FF | 1 | 0.0275±0.0013 | 3.23±0.05 | 121.3±5.3 | 3.08 |
|  |  | 2 | 0.0289±0.0018 | 3.56±0.04 | 127.8±8.6 | 3.25 |
| 16 |  | 4 | 0.0300±0.0006 | 3.88±0.05 | 134.2±3.2 | 3.41 |
|  |  | 0 | 0.0294±0.0014 | 4.15±0.04 | 148.8±7.5 | 3.78 |
|  | BTree | 1 | 0.0314±0.0019 | 4.43±0.10 | 148.7±9.6 | 3.78 |
|  |  | 2 | 0.0329±0.0014 | 4.56±0.05 | 146.2±6.5 | 3.71 |
|  |  | 4 | 0.0335±0.0016 | **4.78**±0.02 | **150.5**±6.6 | **3.82** |

# L. Results on RTX 3090 GPU

## L.1. EvaByte

### L.1.1. Speculative Sampling

*Table 19*

|    |     | ⏱$\mu_{\text{lat}}$ ↓ | ☑$\mu_{\text{acc}}$ ↑ | $\mu_{\text{tok/s}}$ ↑ | $\max_{\text{tok/s}}$ |
|------|-----|--------|------|-------|--------|
| FF | 1 | 0.0548 | 5.14 | 96.2 | 160.66 |
|    | 8 | 0.0591 | 5.64 | 98.0 | 157.02 |
|    | 16 | 0.0577 | 5.73 | 102.3 | 158.35 |
| CP | 32 | 0.0574 | 5.89 | 105.8 | 154.99 |
|    | 64 | 0.0567 | 5.98 | 108.8 | 154.92 |
|    | 128 | 0.0590 | 5.98 | 104.4 | 151.02 |

*Table 20*

|    |    |      | ⏱$\mu_{\text{lat}}$ ↓ | ☑$\mu_{\text{acc}}$ ↑ | $\mu_{\text{tok/s}}$ ↑ | ⏭×$_{\text{STP}}$ |
|----|----|------|--------|------|-------|------|
| 1 | 1 | STP | 0.0472 | — | 21.2 | 1.00 |
|   | 1 | FF | 0.0548 | 5.14 | 96.2 | 4.53 |
| 8 |    | HMM | 0.0657 | 6.03 | 94.3 | 4.44 |
|   | 32 | BTree | 0.0597 | 6.05 | 104.4 | 4.91 |
|   |    | CP | 0.0574 | 5.89 | 105.8 | 4.98 |
|   | 1 | FF | 0.0556 | 5.37 | 99.2 | 4.67 |
| 16 |    | HMM | 0.0774 | 6.88 | 91.2 | 4.29 |
|   | 32 | CP | 0.0594 | 6.18 | 107.1 | 5.04 |
|   |    | BTree | 0.0639 | 6.71 | 108.2 | 5.09 |

*Table 21*

|    |      |   | ⏱$\mu_{\text{lat}}$ ↓ | ☑$\mu_{\text{acc}}$ ↑ | $\mu_{\text{tok/s}}$ ↑ | ⏭×$_{\text{STP}}$ |
|----|------|---|--------|------|-------|------|
| 1 | STP |   | — | 0.0472 | — | 21.2 | 1.00 |
|   | FF | 0 | 0.0553 | 5.13 | 95.0 | 4.47 |
|   |    | 1 | 0.0569 | 5.05 | 90.9 | 4.28 |
|   |    | 2 | 0.0586 | 5.11 | 89.3 | 4.20 |
|   |    | 4 | 0.0624 | 5.12 | 84.1 | 3.96 |
| 8 | BTree | 0 | 0.0595 | 6.04 | 104.5 | 4.92 |
|   |    | 1 | 0.0614 | 6.19 | 104.0 | 4.90 |
|   |    | 2 | 0.0625 | 6.21 | 102.6 | 4.83 |
|   |    | 4 | 0.0667 | 6.22 | 96.3 | 4.53 |
|   | FF | 0 | 0.0554 | 5.35 | 99.1 | 4.67 |
|   |    | 1 | 0.0597 | 5.46 | 94.0 | 4.42 |
|   |    | 2 | 0.0605 | 5.61 | 95.3 | 4.49 |
|   |    | 4 | 0.0635 | 5.64 | 91.4 | 4.30 |
| 16 | BTree | 0 | 0.0620 | 6.85 | 114.0 | 5.37 |
|   |    | 1 | 0.0648 | 7.24 | 115.6 | 5.44 |
|   |    | 2 | 0.0666 | 7.41 | 115.3 | 5.43 |
|   |    | 4 | 0.0697 | 7.50 | 111.7 | 5.26 |

## L.1.2. GREEDY SPECULATIVE DECODING

*Table 22*

|     |     | $\bigcirc \mu_{lat} \downarrow$ | $\checkmark \mu_{acc} \uparrow$ | $\mu_{tok/s} \uparrow$ | $max_{tok/s}$ |
|-----|-----|-------------------|-------------------|---------------|---------------|
| FF  | 1   | 0.0552 | 6.87 | 128.9 | 162.65 |
|     | 8   | 0.0579 | 6.66 | 119.7 | 160.28 |
|     | 16  | 0.0580 | 6.67 | 119.7 | 158.39 |
| CP  | 32  | 0.0581 | 6.65 | 119.2 | 161.22 |
|     | 64  | 0.0584 | 6.65 | 118.7 | 155.42 |
|     | 128 | 0.0584 | 6.64 | 118.7 | 151.29 |

*Table 23*

|    |    |       | $\bigcirc \mu_{lat} \downarrow$ | $\checkmark \mu_{acc} \uparrow$ | $\mu_{tok/s} \uparrow$ | $\blacktriangleright\blacktriangleright\!\mid \times_{STP}$ |
|----|----|-------|-------------------|-------------------|---------------|--------------|
| 1  | 1  | STP   | 0.0466 | — | 21.5 | 1.00 |
|    | 1  | FF    | 0.0552 | 6.87 | 128.9 | 5.98 |
| 8  |    | HMM   | 0.0603 | 6.69 | 115.3 | 5.35 |
|    | 32 | BTree | 0.0589 | 6.66 | 117.7 | 5.46 |
|    |    | CP    | 0.0581 | 6.65 | 119.2 | 5.53 |
|    | 1  | FF    | 0.0550 | 7.71 | 145.7 | 6.76 |
| 16 |    | HMM   | 0.0657 | 8.38 | 133.6 | 6.20 |
|    | 32 | CP    | 0.0602 | 7.70 | 134.3 | 6.23 |
|    |    | BTree | 0.0621 | 8.13 | 137.5 | 6.38 |

*Table 24*

|    |       |   | $\bigcirc \mu_{lat} \downarrow$ | $\checkmark \mu_{acc} \uparrow$ | $\mu_{tok/s} \uparrow$ | $\blacktriangleright\blacktriangleright\!\mid \times_{STP}$ |
|----|-------|---|-------------------|-------------------|---------------|--------------|
| 1  | STP   |   | — | 0.0466 | — | 21.5 | 1.00 |
|    | FF    | 0 | 0.0563 | 6.87 | 126.6 | 5.88 |
|    |       | 1 | 0.0588 | 6.87 | 121.0 | 5.62 |
|    |       | 2 | 0.0598 | 6.88 | 119.5 | 5.55 |
| 8  |       | 4 | 0.0633 | 6.86 | 112.5 | 5.22 |
|    | BTree | 0 | 0.0598 | 6.68 | 116.4 | 5.40 |
|    |       | 1 | 0.0624 | 6.70 | 112.0 | 5.20 |
|    |       | 2 | 0.0652 | 6.69 | 107.0 | 4.97 |
|    |       | 4 | 0.0669 | 6.70 | 104.4 | 4.85 |
|    | FF    | 0 | 0.0569 | 7.70 | 140.8 | 6.54 |
|    |       | 1 | 0.0621 | 8.30 | 139.5 | 6.48 |
|    |       | 2 | 0.0624 | 8.51 | 142.6 | 6.62 |
| 16 |       | 4 | 0.0655 | 8.66 | 138.4 | 6.43 |
|    | BTree | 0 | 0.0636 | 8.22 | 136.0 | 6.31 |
|    |       | 1 | 0.0657 | 8.94 | 143.4 | 6.66 |
|    |       | 2 | 0.0680 | 9.03 | 140.2 | 6.51 |
|    |       | 4 | 0.0709 | 9.10 | 135.5 | 6.29 |

## L.2. Llama 3.2 3B

### L.2.1. SPECULATIVE SAMPLING

*Table 25*

|     |    | $\bigcirc\mu_{\text{lat}}\downarrow$ | $\checkmark\mu_{\text{acc}}\uparrow$ | $\mu_{\text{tok/s}}\uparrow$ | $\max_{\text{tok/s}}$ |
|-----|----|--------|------|------|--------|
| FF  | 1  | 0.0503 | 1.71 | 36.8 | 193.61 |
|     | 8  | 0.0519 | 1.92 | 40.0 | 190.51 |
| CP  | 16 | 0.0524 | 1.96 | 40.6 | 192.18 |
|     | 32 | 0.0517 | 2.04 | 42.7 | 193.86 |

*Table 26*

|    |    |       | $\bigcirc\mu_{\text{lat}}\downarrow$ | $\checkmark\mu_{\text{acc}}\uparrow$ | $\mu_{\text{tok/s}}\uparrow$ | $\blacktriangleright\!\blacktriangleright\!\mid\times_{\text{STP}}$ |
|----|----|-------|--------|------|------|------|
| 1  | 1  | STP   | 0.0410 | —    | 24.5 | 1.00 |
|    | 1  | FF    | 0.0503 | 1.71 | 36.8 | 1.51 |
| 8  |    | HMM   | 0.0605 | 2.21 | 39.8 | 1.63 |
|    | 32 | CP    | 0.0517 | 2.04 | 42.7 | 1.75 |
|    |    | BTree | 0.0554 | 2.19 | 42.9 | 1.76 |
|    | 1  | FF    | 0.0506 | 1.72 | 36.8 | 1.50 |
| 16 |    | HMM   | 0.0703 | 2.25 | 34.8 | 1.42 |
|    | 32 | CP    | 0.0525 | 2.08 | 42.6 | 1.74 |
|    |    | BTree | 0.0555 | 2.38 | 46.0 | 1.88 |

*Table 27*

|    |       |   | $\bigcirc\mu_{\text{lat}}\downarrow$ | $\checkmark\mu_{\text{acc}}\uparrow$ | $\mu_{\text{tok/s}}\uparrow$ | $\blacktriangleright\!\blacktriangleright\!\mid\times_{\text{STP}}$ |
|----|-------|---|--------|------|------|------|
| 1  | STP   | — | 0.0410 | —    | 24.5 | 1.00 |
|    |       | 0 | 0.0497 | 1.71 | 37.4 | 1.53 |
|    | FF    | 1 | 0.0533 | 1.94 | 40.0 | 1.64 |
|    |       | 2 | 0.0538 | 2.05 | 41.8 | 1.71 |
| 8  |       | 4 | 0.0568 | 2.11 | 40.9 | 1.67 |
|    |       | 0 | 0.0540 | 2.36 | 47.1 | 1.93 |
|    | BTree | 1 | 0.0566 | 2.56 | 48.8 | 1.99 |
|    |       | 2 | 0.0577 | 2.68 | 50.1 | 2.05 |
|    |       | 4 | 0.0604 | 2.71 | 48.6 | 1.99 |
|    |       | 0 | 0.0495 | 1.71 | 37.5 | 1.53 |
|    | FF    | 1 | 0.0534 | 1.91 | 39.5 | 1.61 |
|    |       | 2 | 0.0546 | 2.04 | 41.0 | 1.68 |
| 16 |       | 4 | 0.0573 | 2.05 | 39.8 | 1.63 |
|    |       | 0 | 0.0562 | 2.40 | 46.2 | 1.89 |
|    | BTree | 1 | 0.0583 | 2.60 | 48.3 | 1.97 |
|    |       | 2 | 0.0596 | 2.65 | 48.1 | 1.97 |
|    |       | 4 | 0.0620 | 2.74 | 47.7 | 1.95 |

## L.2.2. GREEDY SPECULATIVE DECODING

*Table 28*

|    |    | $\bigcirc \mu_{\text{lat}}\downarrow$ | ☑$\mu_{\text{acc}}\uparrow$ | $\mu_{\text{tok/s}}\uparrow$ | $\max_{\text{tok/s}}$ |
|----|----|--------|------|------|--------|
| FF | 1  | 0.0448 | 2.69 | 61.7 | 195.66 |
|    | 8  | 0.0485 | 3.23 | 70.0 | 193.55 |
| CP | 16 | 0.0496 | 3.41 | 72.5 | 194.39 |
|    | 32 | 0.0496 | 3.45 | 73.6 | 193.76 |

*Table 29*

|    |    |       | $\bigcirc \mu_{\text{lat}}\downarrow$ | ☑$\mu_{\text{acc}}\uparrow$ | $\mu_{\text{tok/s}}\uparrow$ | ⏭$\times_{\text{STP}}$ |
|----|----|-------|--------|------|------|------|
| 1  | 1  | STP   | 0.0413 | —    | 24.3 | 1.00 |
|    | 1  | FF    | 0.0448 | 2.69 | 61.7 | 2.54 |
| 8  |    | CP    | 0.0496 | 3.45 | 73.6 | 3.03 |
|    | 32 | BTree | 0.0506 | 3.72 | 77.6 | 3.20 |
|    |    | HMM   | 0.0519 | 3.90 | 78.9 | 3.25 |
|    | 1  | FF    | 0.0458 | 2.70 | 60.8 | 2.50 |
| 16 |    | CP    | 0.0500 | 3.20 | 68.0 | 2.80 |
|    | 32 | HMM   | 0.0559 | 4.10 | 77.1 | 3.17 |
|    |    | BTree | 0.0518 | 4.00 | 81.7 | 3.36 |

*Table 30*

|    |       |   | $\bigcirc \mu_{\text{lat}}\downarrow$ | ☑$\mu_{\text{acc}}\uparrow$ | $\mu_{\text{tok/s}}\uparrow$ | ⏭$\times_{\text{STP}}$ |
|----|-------|---|--------|------|------|------|
| 1  | STP   |   | —      | 0.0413 | — | 24.3 | 1.00 |
|    | FF    | 0 | 0.0441 | 2.68 | 62.4 | 2.57 |
|    |       | 1 | 0.0477 | 3.25 | 70.6 | 2.91 |
|    |       | 2 | 0.0492 | 3.54 | 74.9 | 3.08 |
| 8  |       | 4 | 0.0524 | 3.89 | 77.3 | 3.18 |
|    | BTree | 0 | 0.0503 | 3.84 | 80.6 | 3.32 |
|    |       | 1 | 0.0524 | 4.11 | 82.6 | 3.40 |
|    |       | 2 | 0.0532 | 4.30 | 85.2 | 3.51 |
|    |       | 4 | 0.0554 | 4.45 | 84.6 | 3.48 |
|    | FF    | 0 | 0.0444 | 2.73 | 63.2 | 2.60 |
|    |       | 1 | 0.0479 | 3.23 | 70.0 | 2.88 |
|    |       | 2 | 0.0495 | 3.55 | 74.4 | 3.06 |
| 16 |       | 4 | 0.0531 | 3.89 | 76.2 | 3.14 |
|    | BTree | 0 | 0.0517 | 4.16 | 85.0 | 3.50 |
|    |       | 1 | 0.0538 | 4.38 | 86.1 | 3.54 |
|    |       | 2 | 0.0549 | 4.61 | 88.6 | 3.65 |
|    |       | 4 | 0.0573 | 4.74 | 87.4 | 3.60 |

## M. Further Plots on Extending the MTP Token Window to $n = 16$

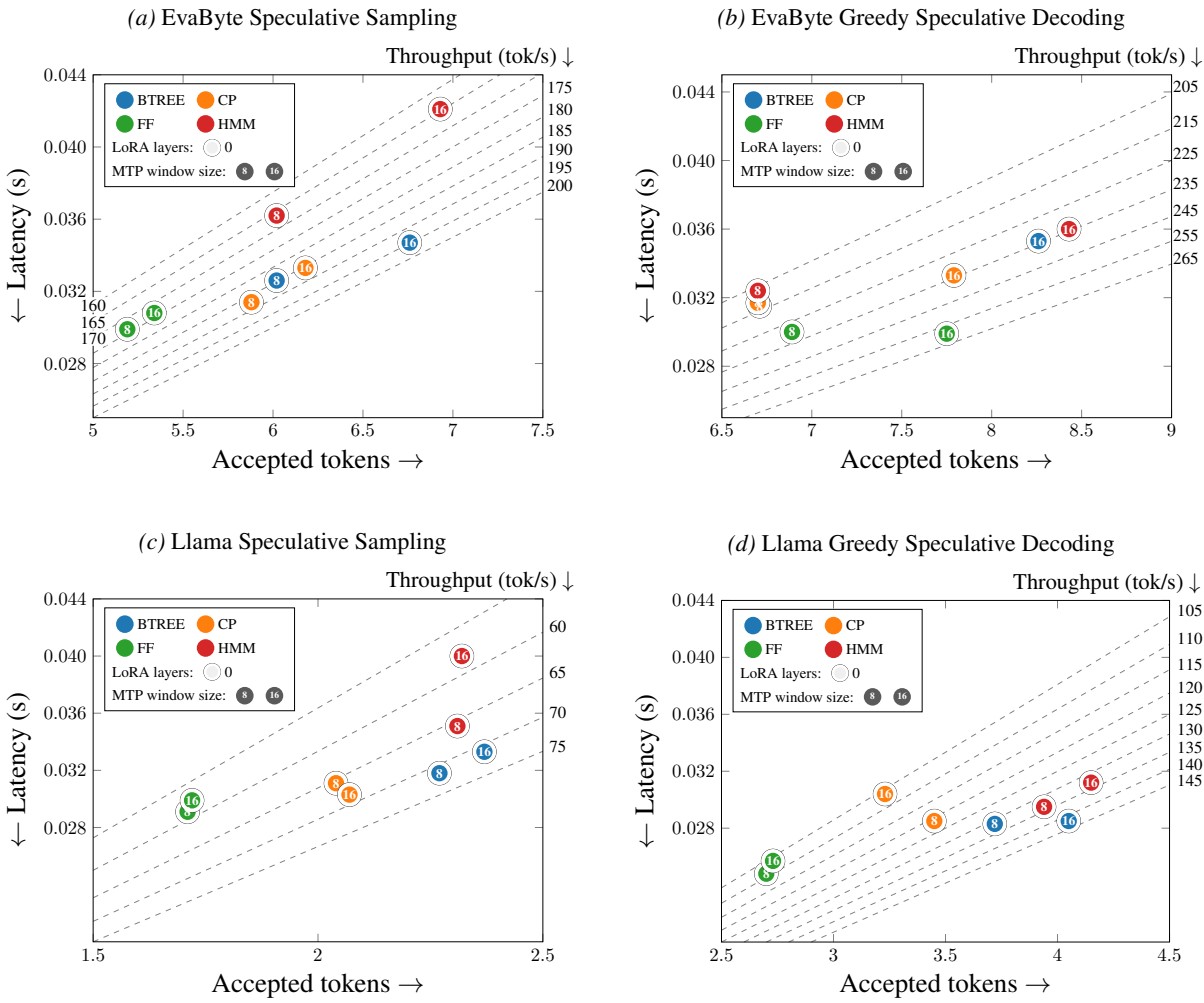

*Figure 6.* Extending the MTP window to $n = 16$ is beneficial for models with no LoRA layers. Marker positions show the mean number of accepted tokens and the mean latency; iso-lines indicate the corresponding ratio of means, which may differ slightly from the per-run mean throughput reported in the tables above. More specifically, extending the MTP window of expressive PCs such as MTPC-HMM and MTPC-BTREE to $n = 16$ boosts their ☑$\mu_{\text{acc}}$ in all settings: HMM 16 is to the right of HMM 8 and BTREE 16 is to the right of BTREE 8 in all four subplots. Importantly, MTPC-BTREE improves $\mu_{\text{tok/s}}$ for all cases apart from Evabyte with greedy speculative decoding (top right), highlighting its strong expressiveness–latency trade-off. We hypothesise that the reason MTPC-FF is so successful for EvaByte in the greedy setting is that this setup aligns with the objective optimised during pretraining because of the independence assumption. In contrast, although MTPC-HMM has a high ☑$\mu_{\text{acc}}$, it lags behind in throughput because of its high latency due to its AR nature (HMM 16 is much higher in the plots than HMM 8).

## N. Further Plots on Adding more LoRA Layers to the Draft Model

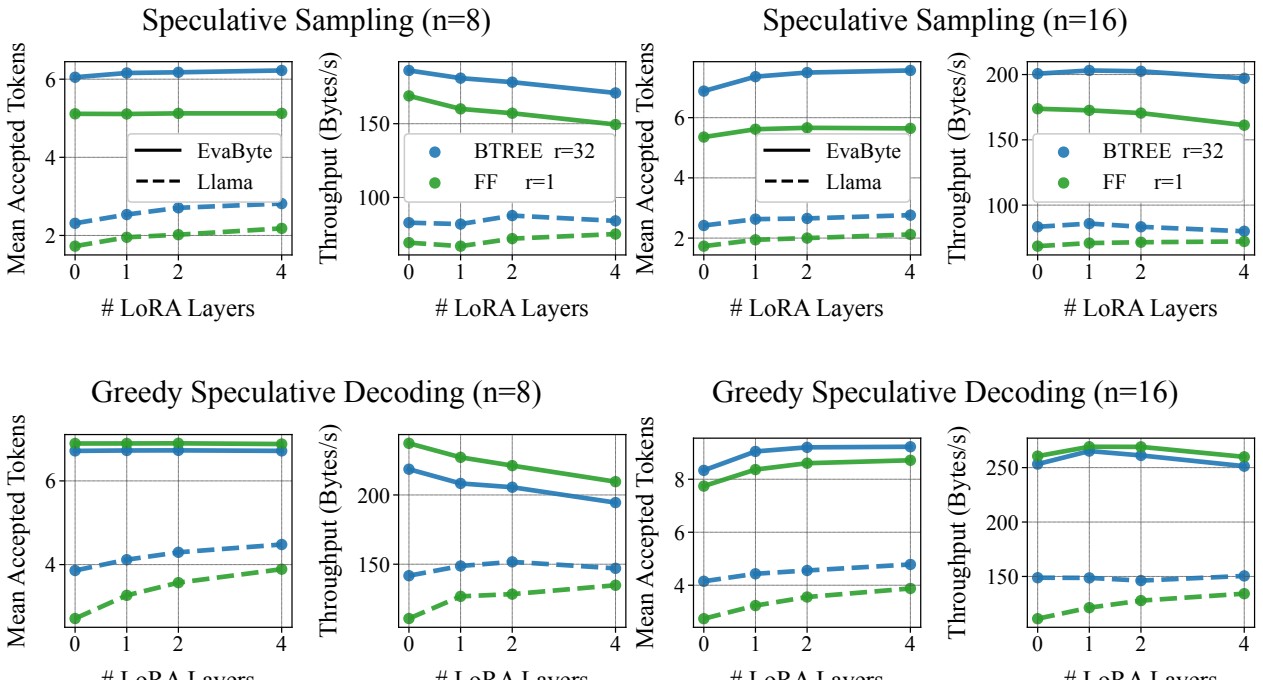

*Figure 7.* Increasing the expressiveness of our draft model by adding LoRA layers. For the $n = 8$ case, we get very little improvement for sampling with EvaByte and no improvement for greedy speculative decoding. We believe this is because EvaByte has been pretrained as a MTP model with $n = 8$ and as such the draft backbone has already converged to MTP representations which cannot be improved further. When we move to $n = 16$, EvaByte acceptance rates are boosted significantly both for speculative sampling and greedy speculative decoding and the optimal throughput is obtained with 1 LoRA layer. On the other hand, the Llama acceptance rates and throughput are boosted in all cases.

