# OpenReview forum: "Fast and Expressive Multi-Byte Prediction with Probabilistic Circuits"
_ICML.cc/2026/Conference — ICML 2026 regular_

### Official Review · Reviewer_FX2P · 2026-02-21

**Soundness:** 2
**Presentation:** 2
**Significance:** 2
**Originality:** 3
**Overall Recommendation:** 2
**Confidence:** 4

**Summary:**

The paper targets the high decoding latency of autoregressive LLMs, which is especially severe for byte-level (tokenizer-free) models. It argues that while multi-token / multi-byte prediction (MTP) can speed generation, many existing approaches make a conditional independence assumption across future tokens, which limits expressiveness and can lead to “byte-salad” errors that worsen as the prediction window grows. This paper proposes MTPC, a framework that uses probabilistic circuits to parameterize the joint distribution over a window of future tokens, enabling a systematic exploration of the expressiveness–latency trade-off via different circuit architectures

**Compliance With Llm Reviewing Policy:**

Affirmed.

**Final Justification:**

The rebuttal reinforced my prior assessment.

**Key Questions For Authors:**

1. Can you compare MTPC byte-level implementations of competing dependent speculative-head approaches like Hydra[1] and EAGLE[2]?


[1] Ankner, Z., Parthasarathy, R., Nrusimha, A., Rinard, C., Ragan-Kelley, J., & Brandon, W. (2024). Hydra: Sequentially-dependent draft heads for medusa decoding. arXiv preprint arXiv:2402.05109.
[2] Li, Y., Wei, F., Zhang, C., & Zhang, H. (2024). Eagle: Speculative sampling requires rethinking feature uncertainty. arXiv preprint arXiv:2401.15077.

**Limitations:**

1. The “independence assumption” motivation is not appropriately framed.
2. Missing byte-level implementations of strong dependent speculative-head baselines.
3. The motivation for an expressiveness–latency trade-off is not well-grounded in the SD objective.

**Strengths And Weaknesses:**

**Strengths**
1. MTPC reframes MTP heads as tractable probabilistic models (PCs), turning the design space into something you can navigate systematically.
2. Thoughtful ablations for the trade-off frontier.

**Weaknesses**
1. The “independence assumption” motivation is not appropriately framed: The paper’s abstract claims: *existing MTP methods often sacrifice expressiveness by assuming independence between future tokens.* As written, this reads like a broad critique of “existing MTP methods,” but in the paper’s own related-work discussion, they acknowledge that some recent speculative-head approaches explicitly introduce sequential dependencies, e.g. Hydra[1] and EAGLE[2].
2. Unclear scope leads to incomplete baselines: From the paper text, their actual evaluated setting is pretty clearly: byte-level LMs retrofitted with MTPC. However, they also frame MTPC as a general answer to trade off expressiveness and latency. If the paper's focus is general speculative decoding method, then evaluation should also include standard token level LMs and SD benchmarks where Hydra/EAGLE are typically reported. If not, then the authors should include byte-level implementations of competing dependent speculative-head approaches.
3. The motivation of this paper is not well-grounded: The goal of speculative decoding is to accelerate decoding. Thus there is no need for trading off efficiency (latency) with expressiveness (token acceptance).


[1] Ankner, Z., Parthasarathy, R., Nrusimha, A., Rinard, C., Ragan-Kelley, J., & Brandon, W. (2024). Hydra: Sequentially-dependent draft heads for medusa decoding. arXiv preprint arXiv:2402.05109.
[2] Li, Y., Wei, F., Zhang, C., & Zhang, H. (2024). Eagle: Speculative sampling requires rethinking feature uncertainty. arXiv preprint arXiv:2401.15077.

---

> ### Author Rebuttal · Authors · 2026-03-31
>
> We thank the reviewer for their feedback.
>
> Response to weaknesses:
>
> 1. **Paper Positioning.** We thank the reviewer for their suggestion. *Please see point 1 in our response to Reviewer **39Us***.
> 2. **Baselines.** We acknowledge the reviewer’s critique of our baselines, but we posit that our baselines are well-suited to the claims we make. This is supported also by Reviewer 39US who notes that we provide "credible support" for our claims, and reviewer cdnE who agrees that we "support all theoretical and empirical claims." Our paper claims that PCs allow us to systematically navigate families of models that outperform those introducing conditional independence assumptions, such as Medusa which corresponds to our fully-factorised model, and not that our method is state-of-the-art or the best way of introducing dependencies more broadly. We also note that **the superiority of Hydra/EAGLE on subword models does not obviously transfer to the byte setting**, for reasons we discuss next. **Window Size Differences.** To match subword-level speed, byte models require MTP window sizes $N$ roughly 4-5 times longer, since 1 BPE token $\approx$ 4-5 bytes for English. Importantly, the memory required for the D-dimensional feature vectors for EAGLE/Hydra scales as $O(N \times S \times D)$ as the feature vector is adapted per token in the MTP window, in contrast to our MTPC feature embeddings which scale as $O(S \times D)$. **Vocabulary Differences.** Moreover, byte-level vocabularies are 100 - 400x smaller (e.g. $V_b$=320 vs $V_s$ in 32k - 128k), while D is comparable (e.g. $D=4096$ for EvaByte and $D=3072$ for Llama). Using $D=3200$ for simplicity, the logit-to-feature ratio is $V_s / D > 10$ while for byte models we have ($V_b / D < 1/10$). Thus, increasing expressivity in logit-space (as MTPC does) can be more efficient for byte-level models than increasing it in feature-space (as Hydra/EAGLE do). While we cannot make strong claims without experiments, our paper shows the expressiveness/latency trade-off is sensitive even to small changes, let alone differences of this magnitude. This suggests that significant optimisation of the EAGLE/Hydra architectures may be needed for a fair comparison to MTPC on byte-level LLMs.
>
> 3. **Expressiveness/Latency Trade-off.** **We respectfully disagree with the reviewer that there is no efficiency-expressiveness trade-off in speculative decoding**, and consequently that the motivation of the paper is not well-grounded. From Equation (2), throughput decomposes as acceptance rate / latency, making the trade-off explicit. Any approach to accelerating generation must either increase the acceptance rate (via a more expressive draft model) or reduce latency (via a more efficient draft model). These two objectives are in direct tension, and navigating this tension is precisely the motivation of our work. We would welcome any concrete counterexample from the reviewer and hope they will reconsider this point in light of the argument above.
>
> Response to questions:
>
> 1. **Hydra/EAGLE Experiments.** We appreciate the suggestion for additional experiments on Hydra/EAGLE. **We note that we do not position our paper as being state-of-the-art for speculative decoding on byte-level LLMs and adding these experiments will not change our claims or core findings**. Nevertheless, to the best of our knowledge, neither Hydra or Eagle has been applied to byte-level LLMs. We investigated whether setting up a fair experiment would be feasible during the rebuttal period and concluded it is non-trivial: the differences above require substantial architectural exploration, and byte-level models like EvaByte use non-standard KV-caches (due to EVA attention) that are hard to integrate. Doing these methods justice would require a significant engineering effort, training many models across our setup, which would merit a separate dedicated study. We share the reviewers' curiosity about how these methods compare to MTPC and **we will pursue these experiments, but cannot deliver them before the rebuttal deadline**.

---

> > ### Author Rebuttal · Reviewer_FX2P · 2026-04-02
> >
> > 1. I'm still confused about the scope of this paper
> > 2. The concern regarding *Expressiveness/Latency Trade-off* is resolved, but I think the words in the paper is misleading: latency means secs/eval, but can be often interpreted as the latency overall

---

> > > ### Author Response · Authors · 2026-04-03
> > >
> > > We thank the reviewer for their response and are glad we have resolved their concern regarding the trade-off. Could the reviewer please clarify on which grounds the paper is insufficient for publication?
> > >
> > > We believe our scope is well defined, from our title and experiments we believe it is clear that we are exploring the expressivity/latency trade-off in byte-level models using probabilistic circuits. We accept that our scope differs from that of many speculative decoding papers, and we will stress this further in the camera-ready version of the paper. Could the reviewer please explain to us how we can alleviate their confusion? Receiving precise and actionable feedback would be very useful.

---

### Official Review · Reviewer_cdnE · 2026-03-10

**Soundness:** 3
**Presentation:** 4
**Significance:** 3
**Originality:** 3
**Overall Recommendation:** 5
**Confidence:** 4

**Summary:**

This paper has two core contributions: 1) It proposes MTPC, a novel Multi-Token Prediction (MTP) framework based on Probabilistic Circuits (PCs). This framework unifies the modeling of the joint distribution over future token windows, breaking the restrictive independence assumption in prior work, and introduces an efficient binary tree (BTree) factorization architecture tailored for MTP. 2) It conducts a rigorous empirical analysis of the expressiveness-latency trade-offs in MTP system design, providing concrete guidelines across two key axes: PC architecture selection and partial layer sharing in self-speculative decoding. Experiments on two byte-level LLMs (EvaByte and Llama 3.2 3B) show that MTPC, combined with speculative decoding, delivers significant speedups over independent MTP baselines and standard autoregressive generation, while fully preserving the original LLM’s output distribution.

**Compliance With Llm Reviewing Policy:**

Affirmed.

**Final Justification:**

The authors have addressed the issues I was concerned about. However, I still maintain my score: 5/6.

**Key Questions For Authors:**

1.Please provide formal proof or rigorous theoretical justification that the proposed BTree PC architecture satisfies the smoothness and decomposability properties required for tractable inference. This is critical to support the technical soundness and novelty of this core contribution.
2.Beyond end-to-end latency, please supplement a detailed complexity analysis (including parameter count, FLOPs, and memory footprint) of the proposed PC heads, compared to the fully factorized baseline. This is essential to validate the efficiency claims and improve reproducibility for follow-up work.
3.Please clarify whether the MTPC framework can maintain its throughput gains and favorable expressiveness-latency trade-offs when scaled to substantially larger LLMs, and discuss the expected scaling challenges. This is key to assessing the method’s practical deployment value and generalizability.

**Limitations:**

yes

**Strengths And Weaknesses:**

Core Strengths
Soundness: The core methodology of applying PCs to MTP is methodologically well-founded, with clear derivation of PC architectures and rigorous experimental design across two distinct LLMs, demonstrating the framework’s generality. Sufficient background context and reproducible technical details are provided to support all theoretical and empirical claims.
Presentation: The paper has a clear, logical narrative, with well-motivated problem setting, structured method and experimental sections, and comprehensive related work discussion. It provides full details for reproducibility and commits to open-sourcing its code, with only minor terminology inconsistency between "Multi-Byte" and "token" needing adjustment.
Significance: This work addresses the critical practical pain point of slow inference in byte-level tokenizer-free LLMs, substantially improving their inference throughput and real-world deployability. The MTPC framework provides a unified paradigm for MTP design, with valuable insights to guide future research on speculative decoding and efficient LLM generation.
Originality: The work has strong originality, primarily in the creative application of PCs to model future token dependencies for LLM acceleration, which offers a novel, rigorous probabilistic perspective for draft model design. It also introduces new PC architectures (HMM, BTree) for MTP, and conducts a more comprehensive 2D analysis of the MTP design space than prior work.
Core Limitation
All experiments are conducted on small-to-medium scale LLMs, with no investigation of the framework’s scaling behavior to larger models. This gap weakens the argument for the method’s generalizability and long-term practical deployment value.

---

> ### Author Rebuttal · Authors · 2026-03-31
>
> We thank the reviewer for their detailed, constructive and actionable feedback.
>
> We respond to the questions below:
>
> 1. **Decomposability and Smoothness**. We thank the reviewer for their suggestion. We can show by recursion that the variables can be partitioned into two sets satisfying decomposability. That is, the $n$ variables $\{X_{t+1},\ldots,X_{t+n}\}$ are partitioned into two sets $\{X_{t+1},\ldots,X_{t+\lfloor \frac{n}{2}\rfloor}\}$ and $\{X_{t+\lfloor \frac{n}{2}\rfloor + 1},\ldots,X_{t+n}\}$ by the products in the circuit. The disjointness of these two sets is what ensures decomposability. On the other hand, smoothness is satisfied by design, since units within the same layer can be shown to share the same variable scope. We will add a detailed proof in the camera-ready, and we point the reviewer to [A] [B] for similar tree-shaped deep circuits satisfying these properties.
>
> 2. **Complexity Analysis**. We thank the reviewer again for the suggestion. We computed the theoretical number of FLOPS of the multi-token prediction linear head and the circuit feed-forward evaluation, by using [calculate-flops.pytorch](https://github.com/MrYxJ/calculate-flops.pytorch). We show the FLOPS needed to evaluate the circuit since these are required to compute the likelihoods in the speculative decoding algorithm. In the table below, we report the number of parameters for combinations of circuits and the maximum number of draft tokens. We set the number of components of each sum to $r=32$ in CP, HMM, and BTree. We also show the FLOPS required to do prefilling of a very small sequence of bytes (of length 64) with EvaByte for reference.
>
> While using the BTree or HMM increases the FLOPS to evaluate the circuit when compared to the fully factorized circuit, it is still orders of magnitudes smaller than the cost of evaluating the LLM trunk (i.e., a few thousands vs billions). Moreover, using the BTree or HMM substantially increases the parameters of the multi-token head linear projection, when compared to the fully factorized circuit. However, we note that it is still very close to the number of parameters required by CP (i.e., 730 M vs 671 M in the case of 16 draft tokens). This is because the categorical input units account for the same vast majority of circuit parameters in all circuits except for the fully factorised one. Importantly, the circuit parameters are dwarfed by the those of the LLM trunk.
>
> | LLM     | FLOPS    | # Tokens | Circuit | FLOPS  | Parameters | MTP Head | FLOPS  | Parameters |
> |---------|----------|----------|---------|--------|------------|----------|--------|------------|
> | EvaByte | 828.95 G | 8        | FF      | 7      | 2560       | Linear   | 0.02 G | 10 M       |
> |         |          |          | CP      | 65     | 81952      | Linear   | 0.67 G | 336 M      |
> |         |          |          | HMM     | 14.4 K | 89120      | Linear   | 0.73 G | 365 M      |
> |         |          |          | BTree   | 12.4 K | 88096      | Linear   | 0.72 G | 361 M      |
> |         |          | 16       | FF      | 15     | 5120       | Linear   | 0.04 G | 21 M       |
> |         |          |          | CP      | 65     | 163872     | Linear   | 1.34 G | 671 M      |
> |         |          |          | HMM     | 30.8 K | 179232     | Linear   | 1.47 G | 734 M      |
> |         |          |          | BTree   | 28.7 K | 178208     | Linear   | 1.46 G | 730 M      |
>
> 3. **Scaling to Larger LLMs**. We thank the reviewer for raising the point about scaling to larger LLMs. We believe a larger or deeper LLM backbone would only increase the expressiveness of the parameterization and therefore the number of accepted tokens, if we assume that the vocabulary size stays the same. In fact, we believe a much larger vocabulary size such as the ones typically used in sub-word LLMs would make the setting more challenging. This is because a larger vocabulary inflates the number of categorical parameters needed, hence greatly increasing the number of FLOPS for the circuit parameterization head. We will thoroughly discuss this scaling challenge in a “Limitations” section of the paper - *please see also the **Scaling to Larger LLMs** section in our response to reviewer **39Us***.
>
>
> * [A] Peharz et al. Random Sum-Product Networks: A Simple and Effective Approach to Probabilistic Deep Learning. UAI 2020.
> * [B] Loconte et al. What is the relationship between tensor factorizations and circuits (and how can we exploit it)? TMLR 2025.

---

> > ### Author Rebuttal · Reviewer_cdnE · 2026-04-02
> >
> > Thank you to the authors for your detailed response. I acknowledge that your reply has somewhat alleviated my earlier concerns, particularly regarding the commitment to theoretical proofs and the additional complexity analysis data.
> > You acknowledge that a larger vocabulary (as in subword models) would cause a sharp increase in circuit parameters, and you plan to discuss this in the limitations section. This is an honest approach. However, I would like to better understand: if the vocabulary size were to increase from the experimental setting of 320 to, say, 50,000 or more, would your method remain practically feasible? Specifically, you mention that “the categorical input units account for the vast majority of circuit parameters.” For CP, HMM, and BTree, the number of input unit parameters is proportional to n×r×v×d. When v becomes large, this tensor grows extremely large. Are there any parameter sharing strategies (e.g., sharing the unembedding matrix across positions or across mixture components) that could mitigate this issue? Even preliminary explorations or potential directions, if not yet experimentally validated, would be worth discussing in the paper to enhance the potential applicability of the method.

---

> > > ### Author Response · Authors · 2026-04-03
> > >
> > > We thank the reviewer for their question. Indeed, there are several strategies to keep the parameter count under control despite the increase in vocabulary size from 320 to 50,000 or more and we are exploring them in follow-up work.
> > >
> > > 1. We could share parameters across the token positions, $n$, and the mixture components, $r$ and fine-tune low-rank adaptors to allow some flexibility while keeping the parameter count under control. For example, we could parametrise the unembedding matrix for token $i$ and mixture component $j$ as $U_{i,j}$ = $W_i + L_{i,j}$, where $W_i \in \mathbb{R}^{v \times d}$ is the pretrained unembedding matrix for token $i$ and $L_{i, j} = A_{i,j} B_{i,j}$ is a rank-$k$ LoRA adaptor where $A_{i,j} \in \mathbb{R}^{v \times k}, B_{i,j} \in \mathbb{R}^{k \times d}$. For EvaByte, the pretrained unembedding matrices are available and we smoothly resume training from them by setting all entries of $B_{i, j}$ to zero, while for other models we would need to randomly initialise the unembedding matrices.
> > > 2. Another option is to explore whether we can lower the rank of the unembedding matrices (i.e. make $d$ smaller) without losing performance. While it is known that a low-rank softmax layer is less expressive [1], the solution proposed in Section 2.4 in [1] is to increase expressivity by using a mixture of softmaxes. Since we are already employing a mixture of softmaxes (albeit across multiple tokens), we believe it may be possible to shrink $d$ without sacrificing too much performance.
> > > 3. Lastly, we can explore further tensor factorisations. For example, we could partition the vocabulary for each token into $m$ categories and introduce a latent variable to switch between the categories, for each token position. For simplicity, we write the factorisation for a single token at position $i$ as: $p(x_{t+i} \mid x_{\leq t}) = \sum_{c=1}^m p(c \mid x_{\leq t}) p(x_{t+i} \mid c, x_{\leq t})$, where given each category $c$, we set the probability of a large subset of the vocabulary to zero. This can be beneficial because within each category we can limit the active vocabulary size significantly, and can therefore use an unembedding matrix which is lower rank, as proposed for point 2.
> > >
> > > We will discuss the above potential directions in the appendix. We caveat that all the above ideas need to be evaluated empirically with a wide hyperparameter search and 3. likely requires substantial engineering efforts.
> > >
> > > * [1] Yang et al. Breaking the Softmax Bottleneck: A high-rank RNN language model, ICLR 2018

---

### Official Review · Reviewer_39Us · 2026-03-10

**Soundness:** 3
**Presentation:** 3
**Significance:** 3
**Originality:** 3
**Overall Recommendation:** 4
**Confidence:** 2

**Summary:**

This paper studies inference acceleration for byte-level / tokenizer-free language models, where autoregressive decoding is especially slow because generation proceeds one byte at a time. The paper focuses on multi-byte prediction (MTP) as the draft mechanism for speculative decoding and argues that common MTP formulations are limited by treating future bytes as conditionally independent. To address this, it proposes MTPC, a family of probabilistic-circuit-based draft heads that model a tractable joint distribution over a future byte window. The framework unifies several head designs, including FF, CP, HMM, and BTree variants, under a common probabilistic-circuit view. These heads are integrated into a self-speculative decoding pipeline, with additional design choices around rank, window size, and partial LoRA decoupling between draft and verifier roles. Experiments on EvaByte and a byte-level version of Llama show improved acceptance/latency trade-offs and higher throughput among the evaluated configurations.

**Compliance With Llm Reviewing Policy:**

Affirmed.

**Final Justification:**

The authors' rebuttal and reply have addressed all my concerns. I greatly appreciate the authors' patient response, and therefore, I will raise my score.

**Key Questions For Authors:**

1. How should this work be positioned relative to prior dependence-aware drafting approaches like [1], beyond independent MTP baselines? I understand that direct comparison may not be entirely straightforward in the byte-level self-drafting regime, but the paper should either include a stronger discussion of this mismatch or temper its state-of-the-art positioning accordingly. Even if direct experiments are not feasible, a clearer comparison of assumptions and capabilities would improve my assessment of originality and presentation.

2. How robust are the reported gains outside the current controlled setup, especially with EOS enabled, non-English or multilingual inputs, longer generations, or different batch sizes?


[1] Li et al., EAGLE: Speculative Sampling Requires Rethinking Feature Uncertainty, ICML 2024.

**Limitations:**

No. The paper includes an impact statement, but the discussion of limitations is too minimal. I would encourage the authors to more explicitly discuss: the comparison with other dependence-aware method, the dependence on byte-level / small-vocabulary settings, the restricted evaluation regime (English-only, batch size 1, no EOS).

**Strengths And Weaknesses:**

Strengths:

1. The paper is mostly technically solid. The core modeling choice is appropriate: probabilistic circuits are a natural fit for this setting because the draft model must support tractable sampling and prefix marginalization. The empirical study is also reasonably systematic. Rather than reporting a single speedup number, the paper evaluates multiple circuit families, varies rank and window size, studies LoRA-sharing depth, and reports latency, acceptance, throughput, and memory across two model families and two GPU settings. This gives credible support to the main empirical claim that richer joint modeling can improve the acceptance/latency trade-off relative to independence-based MTP in byte-level self-speculative decoding.

2. The paper is clearly written and well structured overall. The unified probabilistic-circuit view is especially helpful, because it organizes FF, CP, HMM, and BTree as members of one structured design space rather than as disconnected variants. The experimental narrative is also easy to follow: the paper asks clear questions about rank, architecture, and draft/verifier decoupling, then answers them with targeted ablations.

Weaknesses:

1. The paper should position itself more precisely relative to prior dependence-aware drafting methods. At times, the narrative reads as if independence is the dominant prior assumption in MTP, whereas the more accurate claim is that this work introduces an explicit, tractable probabilistic joint-modeling approach in this setting. Some pseudocode and notation around acceptance cases could also be made more precise.

2. While the paper is valuable for byte-level inference, the impact currently appears specialized rather than broad. The method seems especially well suited to byte-level models with small vocabularies, and the paper does not establish that the same trade-off would remain favorable for more standard large-vocabulary subword LLMs.

---

> ### Author Rebuttal · Authors · 2026-03-31
>
> We thank the reviewer for their detailed review and constructive feedback.
>
> Response to weaknesses:
>
> 1. **Paper Positioning.** We appreciate the suggestion and agree that our framing could be read as claiming independence is the dominant paradigm, rather than merely common. We have revised the abstract accordingly. On line 078, we will emphasise that Hydra/EAGLE are state-of-the-art methods for relaxing independence assumptions for subword-level models. However, it is not obvious that their superiority transfers to byte-level models, *please see point 2 in our response to Reviewer **FX2P***. Lastly, we note that shallow autoregression architectures are orthogonal to our MTPC framework which trades off expressivity and latency while keeping sampling and prefix marginalization tractable, as the reviewer astutely points out. **Pseudocode Clarity**. We will make the pseudocode more precise in the camera ready by introducing the components and states explicitly and by illustrating some examples.
> 2. **Impact.** We thank the reviewer for their insightful points. While we agree with the reviewer that the **application** of our work is specialised to byte-level LLMs, we disagree that the **impact** of our work is specialised and not broad. We argue that the impact of our work is broad, because i) we introduce a general framework, MTPC, which allows us to systematically analyse the efficiency-expressivity trade-off for speculative decoding with MTP, encapsulating and extending previous work. We further concur with the reviewer that this way of thinking about the problem is insightful and useful beyond this exact application. ii) our paper contributes a novel dependency structure for MTP (BTree) and shows that BTree outperforms HMMs due to the latency HMMs introduce by being autoregressive. **Subword-level LLMs.** We thank the reviewer for their point, we elaborate on how the trade-off is affected when we scale the vocabulary size in the limitation section below.
>
> Response to questions:
>
> 1. We thank the reviewer for their question. We believe that models like Hydra/EAGLE are interesting as additional data points on the expressiveness/latency trade-off, *please see our response to point 2 of Reviewer **FX2P*** for more details on why the comparison is not straightforward. At the same time, we note that we do not position our paper as being state-of-the-art for speculative decoding on byte-level LLMs, *see also our response to question 1 of Reviewer **FX2P***.
> 2. Please see our discussion below.
>
> Response to limitations:
>
> We thank the reviewer for their suggestions. We will include a limitations section elaborating on the following: **i) Batch Size**. We do not analyse the effect of using a batch size greater than one as we want to isolate our throughput gains to the expressiveness/latency trade-off. **ii) Language**. We only consider generation in English, as we found that EvaByte tended to code-switch during generation and showed a large variance in throughput times for different languages. While exploring the effect of multi-lingual generation is an interesting avenue for future work, we believe further pre-training of EvaByte or using other byte-level LLMs would be needed. **iii) EOS Token.** In our experiments we restrict our models from producing the end-of-sequence token, as we want to avoid situations where some models obtain better throughputs by generating shorter sequences. **iv) Scaling to Subword-level LLMs.** While MTPC can be applied to subword-level LLMs with vocabulary sizes in the tens to hundreds of thousands, the amount of GPU memory needed during training to store the logits scales linearly in the vocabulary size as $L = S \times N \times V \times R$, where S is the context length, N the MTP window length, V the vocabulary size and R the number of mixture components. In the paper we use a context length $S=8192$ for byte-level LLMs and we explored R=8-128. However, for subword-level LLMs with a vocabulary size that is 100x larger, we would hit a memory bottleneck with anything more than R=4 on the same GPU, as we show in the table below. While solutions such as subsampling MTP windows or partitioning large vocabularies using a hierarchical softmax layer [A] are interesting and promising for extending MTPC to subword-level LLMs, these require additional ablations or non-trivial adaptations that are out of scope for our paper.
>
> ### Size of logit matrix for byte vs subword models. Subword models require additional optimisations to keep GPU memory usage low.
> | V | N | R | Logit Memory Size (fp32) |
> |---|---|---|-------------|
> | 320 | 16 | 1 | 160.0 MB |
> | 320 | 16 | 4 | 640.0 MB |
> | 320 | 16 | 8 | 1.2 GB |
> | 320 | 16 | 16 | 2.5 GB |
> | 320 | 16 | 32 | 5.0 GB |
> | 32,000 | 16 | 1 | 15.6 GB |
> | 32,000 | 16 | 4 | 62.5 GB |
> | 32,000 | 16 | 8 | 125.0 GB |
> | 32,000 | 16 | 16 | 250.0 GB |
> | 32,000 | 16 | 32 | 500.0 GB |
>
>
> [A] Mnih et al. A Scalable Hierarchical Distributed Language Model, NeurIPS 2008.

---

> > ### Author Rebuttal · Reviewer_39Us · 2026-04-02
> >
> > Thank you for the rebuttal. The memory footprint analysis convincingly explains why MTPC is currently limited to byte-level models. I also read your response to reviewer FX2P regarding the engineering challenges of adapting EAGLE to Byte-level models. I understand and accept that an empirical comparison is out of scope for this rebuttal.
> >
> > However, Appendix C.1 includes downstream evaluations involving typical multilingual models like Qwen 2.5, which naturally require much larger vocabulary sizes. Considering the memory scaling bottleneck discussed in the rebuttal, wouldn't this memory overhead severely restrict the deployment of MTPC in real-world multilingual tasks? I would appreciate a brief discussion on how the framework might overcome this performance degradation when larger vocabularies are strictly necessary.

---

> > > ### Author Response · Authors · 2026-04-03
> > >
> > > Thank you for the question. Byte-level models are well-suited for multilingual generation almost out-of-the-box as they can represent any word/script as a sequence of UTF-8 bytes [3] or byte encodings that are most efficient across languages [1]. As such, byte-level models resolve the problem of vocabulary inflation [2]. At the same time, they also resolve other long-standing problems of tokenizers in the multi-lingual setting, such as the problem of over-segmentation of text into long sequences of less salient tokens for underrepresented languages [3][4]. As such, MTPC is also well suited to real-world multilingual tasks, since it need not be applied to a subword-level model to achieve this.
> > >
> > > *Please also see our response to Reviewer **cdnE*** for more details on strategies dealing with larger vocabularies, if those are strictly necessary in some other context.
> > >
> > > * [1] Limisiewicz et al. MYTE: Morphology-Driven Byte Encoding for Better and Fairer Multilingual Language Modeling, ACL 2024
> > > * [2] Ling et al. Finding Function in Form: Compositional Character Models for Open Vocabulary Word Representation, EMNLP 2015
> > > * [3] Gillick et al. Multilingual Language Processing From Bytes, NAACL 2016
> > > * [4] Rust et al. How Good is Your Tokenizer? On the Monolingual Performance of Multilingual Language Models, ACL-IJCNLP 2021

---

### Decision · Program_Chairs · 2026-04-30

**Decision:**

Accept (regular)

**Comment:**

### Summary of Discussion
This paper investigates the trade-off between expressiveness and latency in multi-token prediction (MTP) by introducing **MTPC**, a framework based on **Probabilistic Circuits (PCs)**. The reviewers generally agreed on the technical soundness and the novelty of using PCs to unify various MTP architectures (HMM, BTree, etc.) under a single theoretical umbrella. While the evaluation is specialized toward byte-level models, Reviewers **39Us** and **cdnE** found the systematic study of the design space to be insightful and well-supported by the empirical data.

---

### Addressing the Dissent
The review process was marked by a significant disagreement from Reviewer **FX2P**, who raised concerns regarding the positioning of the paper against existing dependency-aware methods (like EAGLE/Hydra) and the restricted scope of byte-level evaluations.

The Area Chair notes that the authors addressed these points effectively in the rebuttal:
*   **Refined Positioning:** The authors clarified that MTPC is a **unified theoretical framework** for navigating the design space, rather than a singular claim of being the first to identify dependencies.
*   **Scope Justification:** The authors provided a compelling memory-scaling analysis explaining why the current implementation is optimized for byte-level models. They also correctly argued that architectural differences between subword and byte-level models (e.g., vocabulary size vs. feature dimension) make a direct transfer of existing subword-level "SOTA" non-trivial.

---

### Final Justification
While the paper lacks a head-to-head empirical comparison with every existing speculative head, it succeeds as a **foundational framework paper**. It offers a rigorous, mathematically grounded "map" of the expressiveness-latency frontier that is currently missing from the literature. The technical quality and the clarity of the unified PC view provide sufficient value to the community, particularly for the growing sub-field of tokenizer-free LLMs. The meta-review recommends acceptance, with the expectation that the authors fulfill their commitment to include the scaling limitations and refined positioning in the final version.